

# Ozone (O₃) observations in Saxony, Germany for 1997 - 2020: Trends, modelling and implications for O₃ control

Yaru Wang[1], Dominik van Pinxteren[1], Andreas Tilgner[1], Erik Hans Hoffmann[1], Max Hell[1], Susanne Bastian[2], Hartmut Herrmann[1*]

[1]Atmospheric Chemistry Department (ACD), Leibniz Institute for Tropospheric Research (TROPOS), Permoserstr. 15, Leipzig, 04318, Germany
[2]Saxon State Office for the Environment, Agriculture, and Geology (LfULG), Pillnitzer Platz 3, Dresden Pillnitz, 01326, Germany

*Correspondence to*: Hartmut Herrmann (herrmann@tropos.de)

**Abstract.** Given its importance for human health, vegetation, and the climate, the trends of ground-level ozone (O₃) concentrations in eastern Germany were systematically analysed making use of the long-term O₃ data from 16 measurement stations. The findings indicate that despite reductions in NOₓ concentrations across all sites, O₃ pollution in Saxony has in fact worsened over the past 10 years, especially in densely populated urban areas. The strongest O₃ trend is observed at a traffic-dominated station, with an annual ozone increase of 1.2 µg m$^{-3}$ year$^{-1}$ (or 3.5 % year$^{-1}$), while urban and rural background

stations show more moderate rises, of, on average, 0.5 µg m$^{-3}$ year$^{-1}$ (or 1.1 % year$^{-1}$) over the last decade.

To diagnose O₃ formation and the controlling effects of NOₓ and VOCs over the past decades in this target region, for the first time, detailed photochemical box modelling was performed by means of the complex MCM (Master Chemical Mechanism). Analysis of isopleth diagrams for two seasons indicates that O₃ formation was predominantly VOC-limited at traffic and urban sites from 2000 to 2019. The observed rise in O₃ levels suggests that current efforts to reduce total non-methane volatile organic

compound (TNMVOC, including NMVOCs and oxygenated VOCs) emissions and NOₓ from various sources unfortunately remain insufficient. Based on anthropogenic and biogenic emission data, we recommend that continued NOₓ abatement and further additional VOCs controls, with a focus on solvent use, be implemented in densely populated areas to mitigate O₃ pollution in the coming years.

## 1 Introduction

Tropospheric ground-level ozone, acting simultaneously as a key oxidant and a greenhouse gas, has adverse effects on human health, vegetation such as forests and agricultural crops, and the Earth's climate as a short-lived climate forcer or SLCF (Lefohn et al., 2018; Agathokleous et al., 2020). Apart from typical climate conditions causing the intrusion of stratospheric O₃ from high elevations (Lin et al., 2015; Wang et al., 2020) and long-range atmospheric transport from polluted places (Derwent and Parrish, 2022; Mathur et al., 2022), primarily emitted nitrogen oxides (NOₓ), carbon monoxide (CO), and volatile organic



hydrocarbons (VOCs) are the key precursors, which form $O_3$ in a complex photochemical reaction system depending on the prevailing chemical regime (Crutzen, 1973; Seinfeld and Pandis, 1998, 2016).

The highly nonlinear $O_3$ chemical formation has always been the biggest challenge in controlling ozone pollution. It is widely acknowledged that $O_3$ formation can be limited by VOCs or $NO_x$ or coupling-limited by both VOCs and $NO_x$ (Seinfeld and Pandis, 2016). In the "$NO_x$-limited" regime, reductions in $NO_x$ emissions lead to the most effective of $O_3$ reduction.

Conversely, in the "VOC-limited" regime, reductions in VOCs emissions serve the greatest reduction of $O_3$ pollution, while reductions in $NO_x$ actually increase $O_3$ formation rates. In densely populated European metropolitan areas (e.g. Milan, Athens, Berlin, and Paris) with high $NO_x$ emissions, $O_3$ production tends to be VOC-limited or in a transitional regime whereas in rural and other background areas (such as mountain and ocean sites) it is typically under $NO_x$-limited regimes (Hammer et al., 2002; Gabusi and Volta, 2005; Bossioli et al., 2007; Deguillaume et al., 2008; Beekmann and Vautard, 2010; Melkonyan and Kuttler,

2012; Mar et al., 2016; Feldner et al., 2022).

Europe has launched the Gothenburg Protocol of 1999 to limit emissions from fossil fuel combustion associated with motor vehicles and power plants, resulting in emission reductions, compared to 1990, of 63% ($NO_x$) and 59% (NMVOCs) in 2021 (EEA, 2023). It should be noted that the decline in VOCs has plateaued since 2010, unlike the continued reduction in $NO_x$.

Continuously reduced emissions led to successful mitigation of peak $O_3$ pollution, as reflected in decreasing or stagnant peak

$O_3$ levels across most of ground-based observations (Paoletti et al., 2014; Derwent et al., 2018; Fleming et al., 2018; Yan et al., 2018; Boleti et al., 2019; Ronan et al., 2020), except at remote, high-altitude sites due to the dominant influence of hemispheric background ozone. (Gilge et al., 2010; Boleti et al., 2019).

Despite decreasing peak levels, $O_3$ mean concentrations show opposite and often increasing trends over the last nearly 30 years across most traffic, suburban, and urban, as well as some rural sites (Salthammer et al., 2018; Yan et al., 2018; Diaz et al.,

2020). In the recent 10 - 15 years, stronger $O_3$ increases for certain urban areas as compared to rural sites have been identified (Salthammer et al., 2018; Sicard et al., 2020; Sicard, 2021). At remote or alpine background sites, mean $O_3$ levels have remained stagnant or have shown only slight increases since 2000 (Cristofanelli and Bonasoni, 2009; Parrish et al., 2012; Cooper et al., 2014; Cooper et al., 2020).

Following the clear main trend, many studies have been directed at exploring the causes of rising $O_3$ levels in different

geographic areas, which were attributed to different possible reasons, including reduction of anthropogenic emissions of $NO_x$ and therefore a weakening of the NO titration effect (Sicard et al., 2020; Sicard, 2021), higher biogenic VOCs emission (Curci et al., 2009; Bonn et al., 2018; von Schneidemesser et al., 2018), emissions from land transport sector (involving road traffic, inland navigation and trains) (Mertens et al., 2020), specific synoptic meteorological condition in favour of horizontal downwind transport and vertical transport (Huszar et al., 2016; Kalabokas et al., 2017), effects from increasing temperature

(Melkonyan and Wagner, 2013) or even extreme weather such as heat waves (Yan et al., 2018), higher $CH_4$ emission from increased frequency of biomass burning (Derwent et al., 2007; Cape, 2008), the net impacts of climate change leading to a so-called climate penalty (Colette et al., 2015a; Lin et al., 2020; Otero et al., 2021), or the increasing background $O_3$ levels through long-range transport from polluted areas and increased emissions in Asia (Jenkin, 2008; Derwent et al., 2015; Gaudel et al.,





2018; Mertens et al., 2020). Although extensive efforts have been made to reveal possible reasons for increasing $O_3$ trends, the

complex relations between $O_3$ formation and its drivers in different geographic areas are still not fully understood.

For Germany, a number of studies on long-term (>10 years) changes in ground-level $O_3$ across different geographic areas have been done over past decades (Melkonyan and Kuttler, 2012; Eghdami et al., 2020; Gebhardt et al., 2021). Most studies focus on discussing $O_3$ trends in Germany over a specific period, with few comparing mean trends over the past 10, 20, or even 30 years and explaining how ozone has changed in response to variations in anthropogenic emissions. Generally, and to a certain

extent surprisingly, available annual mean ground-level $O_3$ observations have largely remained stable since 2000 (Cooper et al., 2014; Salthammer et al., 2018; Eghdami et al., 2020) despite the continuous reduction of $O_3$ precursors. For the trends of daily $O_3$ in urban Germany, Sicard et al. (2020) observed a decrease in the period spanning 2005 - 2010 and an increase in 2010 - 2018 and pointed out that the insufficient or inappropriate reduction of anthropogenic emissions had shifted German cities from the $NO_x$-limited to VOC-limited chemistry depending on the ratio of VOCs to $NO_x$. Thus, a rising trend for $O_3$ over

the last decade might have resulted from the lack of significant further reductions in VOCs emissions. However, this finding and attribution is in contrast to other studies which determined the $O_3$ precursor sensitivity with different methods (Ehlers et al., 2016; Otero et al., 2021). For example, in contrast to the conclusion from Sicard et al. (2020), Otero et al. (2021) defined a slope of ozone-temperature relationship based on Generalized Additive Models (GAMs) to analyse the $O_3$ precursor sensitivity using summertime observations of $O_3$ and $NO_x$ from two periods (1999 - 2008 and 2009 - 2018) and concluded that

a great number of the German stations including urban and rural areas showed at all summer temperatures a tendency to move to a $NO_x$-limited chemistry with time. Ehlers et al. (2016) modelled local ozone production rates as a function of OH reactivity of VOCs and $NO_2$ to reveal that under typical summer time conditions, German city sites were located in the VOC-limited regime from 1994 - 2014 and they pointed out that the modelled strong reduction of local ozone production was derived mainly from a much slower reduction of traffic $NO_x$ as compared to VOCs emissions.

Regarding ozone exposure as $O_3$ impacts on human health and vegetation, Sicard et al. (2021) reported the EU-28 urban population was still exposed to $O_3$ levels widely exceeding the WHO limit values for the protection of human health from 2000 to 2017, despite the significant reductions of emissions. The growing risk of potential $O_3$ damage due to increasing stomatal ozone can affect both forest as well as food plants. Regarding forests, data from 2000 - 2014 have been investigated by Proietti et al. (2021), and so far in Europe, and even more seriously in Central Europe, the target value (5000 ppb) for the

protection of vegetation has been exceeded as the AOT40 value, an accumulative dose over a threshold value of 40 ppb.

Overall, chronic $O_3$ levels in Germany continue to be a challenge in terms of air quality and impacts on vegetation and human health. Therefore, the present study focuses on the observed trends of $O_3$ concentrations at 16 measurement stations in the federal state of Saxony in eastern Germany. It contrasts the mean trends of different station types, different concentration levels and different seasons in three time periods and then applies air parcel photochemical modelling to comprehensively evaluate

the efficiency of precursor controls on the observed Saxony $O_3$ trend over the past decades. All in all, the present comprehensive analysis aims to aid $O_3$ pollution control policy in the coming years in Germany.



## 2 Methods and data availability

### 2.1 O₃ and other pollutant observations in Saxony

Hourly values of ground-level $O_3$ concentration between 1997 and 2020 were provided by the air quality monitoring network
of the Saxon State Office for the Environment, Agriculture and Geology (LfULG). Figure 1 shows the location of the
measuring stations, colour-coded according to traffic stations, stations in the urban or rural background, and stations on the
ridge of the Ore Mountains, situated at altitudes between 785 and 1214 m asl. Stations with less than 10 years of ozone data
were excluded and are not shown in Fig. 1. Data analysis was done for 16 stations, including 1 traffic station, 7 urban
background stations, 4 rural background stations and 4 stations on the Ore Mountains ridge. The temporal data coverage at the
station varies, as some of them were opened after 1997 only. The exact periods of $O_3$ measurements for each station in the air
quality monitoring network are shown in Table S1. Data availability within the measurement periods was very good at all
stations, with missing fractions < 5 %.

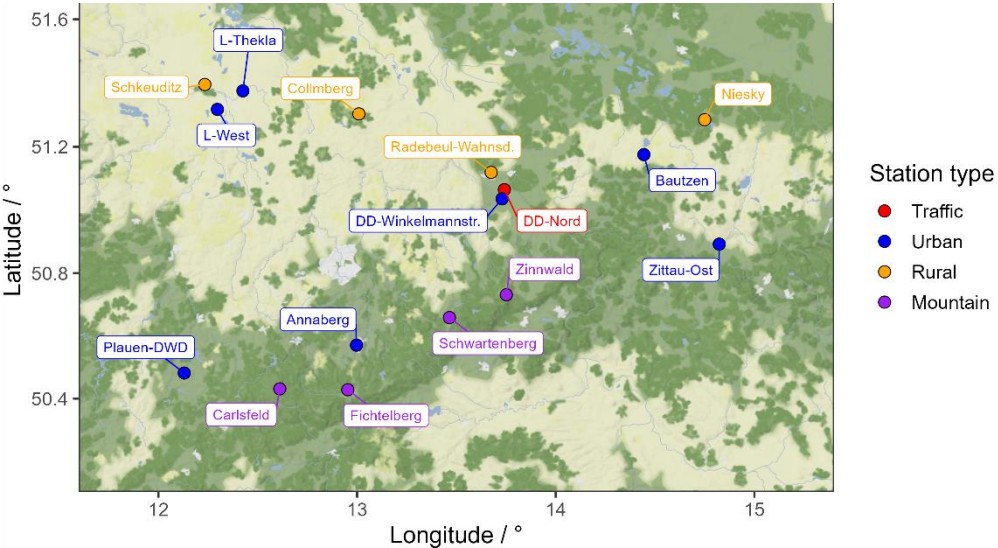

**Figure 1: Map of ozone measuring stations in the Saxony air quality monitoring network that provided data for the present study.
Map produced using the ggmap package (v4.0.0; Kahle and Wickham, 2013) in R (R core team, 2020) with contributions from ©
Stadia Maps © Stamen Design © OpenMapTiles © OpenStreetMap.**

In addition to the hourly $O_3$ values, the concentrations of other air pollutants, NO, $NO_2$ and $NO_x$ were provided as well. The
data available at the respective station are summarised in Table S2, which include nitrogen oxides (NO, $NO_2$ and $NO_x$) and the
meteorological parameters temperature, global radiation, relative humidity, wind direction, wind speed and air pressure.

### 2.2 Trend Analysis



To determine a robust linear trend of $O_3$ concentrations, the non-parametric Theil-Sen estimator according to Sen (1968) was used, which requires no prior assumptions about the statistical distribution of the data and is resistant to outliers. For the calculation, the Theil-Sen function of the package *openair* (Carslaw and Ropkins, 2012) in R (R core Team, 2020) was used, which calculates a straight line and its slope between all points in the data. The median of the slopes of all straight lines then represents the linear trend of the data. The trends were calculated based on monthly averages per station. To ignore the typical annual variation in $O_3$ concentration, a seasonal trend decomposition using a locally weighted scatterplot smoothing (LOESS) function is performed within the *openair* function before the trend calculation when monthly means are used. Furthermore, the *openair* Theil-Sen function derives p-values and uncertainties by bootstrap simulations. The statistical significance of the calculated trend is represented with symbols as follows: $p < 0.001$ = ***, $p < 0.01$ = **, $p < 0.05$ = * and $p < 0.1$ = +. In this paper, all trends with a significance level (probability of error) of 5%, i.e. from $p < 0.05$, which corresponds to at least one asterisk, are considered statistically significant.

## 2.3 Photochemical model simulations

To understand the role of photochemistry for $O_3$ concentration evolution in Saxony, photochemical simulations were performed with the air parcel model SPACCIM (SPectral Aerosol Cloud Chemistry Interaction Model). SPACCIM contains a multiphase chemical model in which the detailed near-explicit gas-phase MCM (Master Chemical Mechanism) is coupled with the aqueous-phase chemical mechanisms CAPRAM (Chemical Aqueous Phase RAdical Mechanism). Detailed descriptions of the SPACCIM model framework can be found in Wolke et al. (2005). In the present study, only the gas-phase chemistry mechanism, MCM-v3.3.1, is used, which comprises 17224 reactions (http://mcm.york.ac.uk/MCM/).

Simulations with two sets of meteorological scenarios were performed, i.e. (i) summer (June, July and August) and (ii) winter (December, January, and February) conditions. All scenarios were driven by anthropogenic and biogenic emission values, meteorological conditions, initial concentrations and deposition rates, respectively.

The summer and winter emission data was based on anthropogenic and biogenic emission inventories in 2019 from the German Environment Agency (UBA) for Germany and Thürkow et al. (2024), and derived for the whole Saxony area (1.06° × 1.76° nested simulation over a domain of 50.9° N latitude and 14.3° E longitude) (see Fig. S1). Besides emission values, other initial parameters, e.g. temperature, photolysis rates, etc., had to be adjusted to their typical daytime and nighttime levels under rural conditions (see Table S3 for details). For trace gas concentrations (except $SO_2$, HONO and PAN) and meteorological parameters, data was derived from the station measurements in Sect. 2.1. The initial $SO_2$ concentrations in summer and winter were obtained from UBA (https://www.umweltbundesamt.de/daten). CO and $CH_4$ were set with 178.1 and 1155.8 µg m$^{-3}$, respectively, referred the measured data from Schaefer (2019) and Zellweger et al. (2009). HONO and PAN concentrations that were not measured at any site were set with the same mixing ratio (0.5 ppb), based on the measurement mean values in Stieger et al. (2018) and Pandey Deolal et al. (2014), respectively. An important aspect of the simulations was to constrain the dry deposition rates of gases, which depends strongly on vegetation cover (Clifton et al., 2020). Considering the different seasonal vegetation covers, different dry deposition velocities (in cm s$^{-1}$) were considered for $NO_2$, $N_2O_5$ and $O_3$ during summer



(NO$_2$ of 0.3 cm s$^{-1}$ (Rondón et al., 1993), N$_2$O$_5$ of 100 cm s$^{-1}$ (considering dry deposition (Zhu et al., 2020) and aerosol uptake) and O$_3$ of 0.8 cm s$^{-1}$ (Clifton et al., 2020), and winter (NO$_2$ of 3 cm s$^{-1}$, N$_2$O$_5$ of 2.0 cm s$^{-1}$, and O$_3$ of 0.08 cm s$^{-1}$) conditions, respectively. Dry deposition rates of other various inorganic gases, such as peroxy acetyl nitrates (PANs), peroxides, carbonyls and acids were kept same in both seasonal simulations and are based on the previous urban or rural scenario setup of CAPRAM

initializations (Wu et al., 2012; Hoffmann et al., 2019; Zhu et al., 2020). It should be noted that different boundary layer heights (BLHs) were assumed for the calculation of the respective emission value. The BLHs in the simulations were set to 500 m at night and 2000 m during daytime in summer and to 250 m at night and 1000 m during daytime in winter. The detailed initial input data (including meteorological conditions, deposition rates, BLHs, chemical initial gas-phase concentrations, e.g. NO$_2$, O$_3$, SO$_2$, CO, etc.) for the two seasonal cases are summarised in Table S3 in the Supporting Information.

To initialise the model as comprehensively as possible, a pre-run for each seasonal scenario was performed. Each pre-run simulation was run for 10 days to have an almost steady-state system with stabilized intermediate products (e.g. radicals (OH, HO$_2$, etc.) and oxygenated VOCs (OVOCs) and other VOCs (e.g. alkanes and chlorinated VOC) concentrations that were not measured at the sites of this study. The initial time of both runs was set to 00:00 Central European Time (CET) on 14 July 2019 (summer case) and 14 January 2019 (winter case), respectively. The output concentrations from the last day of the pre-

run simulations were used as the initial and boundary chemical data for the main simulation (final dominant input concentrations are given in the SI in Table S4). Moreover, the results of final 24-hour summer and winter scenario simulations (defined as the base case simulations) were used to compare with the measured average diurnal patterns of O$_3$, NO and NO$_x$ in both seasons at rural background sites to assess the performance of the model simulations. The good agreement of the base case simulations with measurements (see Sect. 3.3 for details) indicated the model configuration was able to accurately describe

the sensitivities of O$_3$ photochemistry towards its different precursors and impact factors.

To gain a deeper understanding of the O$_3$ trend in the studied area over the past two decades, further photochemical sensitivity simulations were used to derive seasonal isopleth plots of the O$_3$ formation rates under typical summer and winter conditions in Saxony. These isopleths should help derive a more effective O$_3$ control strategy by examining O$_3$ photochemical production against varying levels of NO$_x$ and TNMVOC (including NMVOCs and OVOCs) emission across station types. The sensitivity

simulations were done by scaling the base case emissions of TNMVOC and NO$_x$ 40 times each (Table S5) to achieve a sensible range of resulting TNMVOC and NO$_x$ concentrations in the total 1600 (40 x 40) model runs. The averaged instantaneous rate of net ozone production (NetPO$_3$) during noon time of 12:00 - 13:00 CET for each simulated scenario in both summer and winter conditions was determined for each run. The meteorological conditions and settings for all sensitivity simulations in both summer and winter were identical to those of the base case simulations for each respective season. Upon completion of

the simulations, the isopleth plots were generated by interpolating the resulting formation rates to a regular grid in the TNMVOC vs. NO$_x$ space and then fitting the O$_3$ isopleths to it. The plots illustrate the NetPO$_3$ in relation to the combined ambient concentrations of NO$_x$ and TNMVOC. Subsequently, leveraging data on NO$_x$ and TNMVOC emissions spanning the past two decades (2000, 2005, 2010, 2015, and 2019), we were able to track changes in O$_3$ formation over this period by




examining variations in seasonal isopleths among different station types. This allowed for a comprehensive assessment of the effectiveness of precursor controls in Saxony over time.

## 3 Results

Knowledge of $O_3$ concentrations and trends is important for assessing the effectiveness of existing air pollution control measures and for providing clues for formulating more appropriate control measures in the future. Based on this, first of all, the longer-term changes in the concentration of ground-level $O_3$ at the stations of the Saxony air quality measurement network are examined and presented. Secondly, the mean trends of ozone concentrations in view of different station types, different concentration levels and different seasons are contrasted in three time periods: i) during the entire period of available measurement data, i.e. from 1997 or later, ii) during the 15 years from 2006 to 2020, and iii) during the more recent 10 years from 2011 to 2020. Finally, and third, photochemical modelling was performed and shifts in $O_3$ formation regimes in different station types were then attributed to changing emissions over the past 20 years.

### 3.1 Ozone concentrations and trends

#### 3.1.1 $O_3$ concentrations

Figure 2 shows the distribution of $O_3$ concentrations at the individual monitoring stations. The highest concentrations were observed on the ore mountain ridge. The mean range of $O_3$ concentrations on the mountain ridge were from 69 µg m$^{-3}$ at Schwartenberg (785 m asl) to 78 µg m$^{-3}$ at Fichtelberg (1214 m asl). The highest single hourly value to date was also seen with 282 µg m$^{-3}$ at Schwartenberg. The lowest mean concentrations were observed at the one traffic station in the data set (Dresden-Nord, hereinafter referred to as DD-Nord), with 32 µg m$^{-3}$. Urban background stations showed slightly higher means, depending on the station, approximately between 40 and 55 µg m$^{-3}$. In the rural background, one station closer to the city Schkeuditz, showed a somewhat lower mean concentration level, while the other three stations, Collmberg and Niesky, showed slightly higher values of approximately 55-60 µg m$^{-3}$.





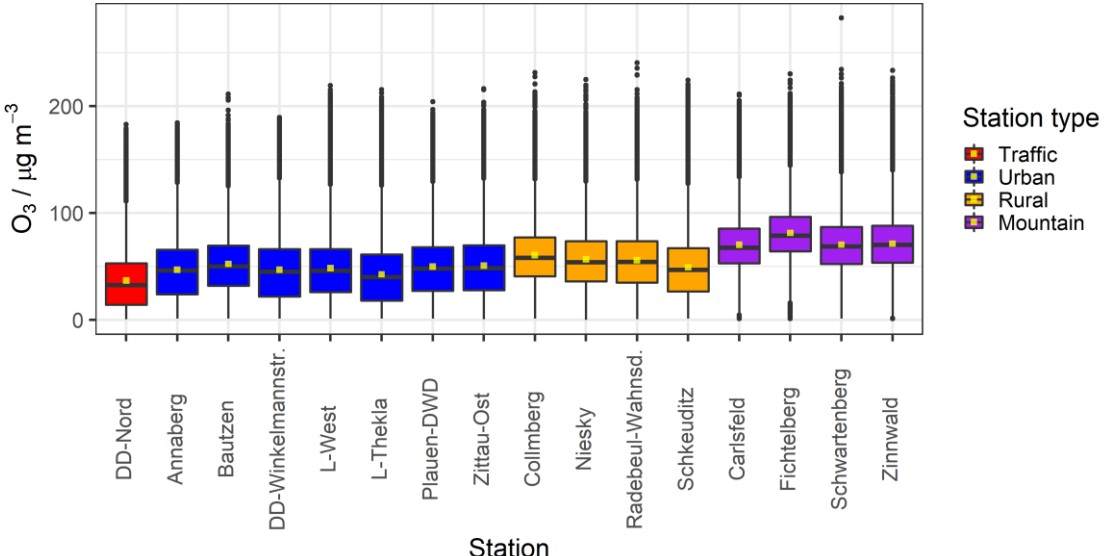

**Figure 2: Boxplots of O₃ concentrations at individual monitoring stations, coloured according to their station type. The middle yellow point and the black horizontal line indicate the mean and median, respectively. The ends of the box the lower and upper quartiles, the antennas the 1.5-fold interquartile range (IQR) and individual points extreme values outside the IQR. Data per station from 1997 onward to 2020. The detailed station data are summarised in Table S1 in the Supporting Information.**

210

In Fig. 2, the trend of O₃ mean concentrations, which tends to increase from traffic stations towards the ore mountains ridge, shows the regional character of O₃. On the one hand, O₃ was formed in the regional background from anthropogenic and biogenic precursor substances, on the other hand, is degraded by reaction with NO close to NO$_x$ sources such as traffic sites. At ore mountain ridge (>780 m), increased mixing concentration from the free troposphere can also play a role. In general, tropospheric O₃ generally shows increasing concentrations with increasing altitude (Davies and Schuepbach, 1994; Cristofanelli and Bonasoni, 2009; Petetin et al., 2016; Li et al., 2018). This is not only due to stronger mixing from higher layers but also due to lower sink strengths, e.g. the reaction with NO or lower deposition fluxes.

### 3.1.2 O₃ trends

Generally, the trends in O₃ concentrations do not necessarily change monotonically over different long periods of time and also at many of the studied sites, shorter periods of increasing, stagnant or decreasing values were seen (Fig. S2). The magnitudes of linear trend estimates therefore depend on the exact time period considered. In addition, the comparison of mean linear trends across several stations is often complicated by differences in data coverage at the respective stations. Therefore, several Theil-Sen trend calculations were carried out for all stations in three different time periods: i) during the entire period

of available measurement data, i.e. from 1997 or later, ii) during the 15 years from 2006 to 2020 (with the exception of the DD-Winkelmannstr. station, where O₃ measurements only started from 2008), and iii) during the more recent 10 years from 2011 to 2020. The reasons for selecting these periods were that, on the one hand, O₃ measurements at almost all stations were



available from 1997 at the earliest, and that, on the other hand, the trends of more recent years may reflect future trends better than longer-term trends including earlier decades. Detailed information on trends of the individual stations is shown in Table S6 for each time period in absolute and relative values and all trends are summarised in Fig. 3.

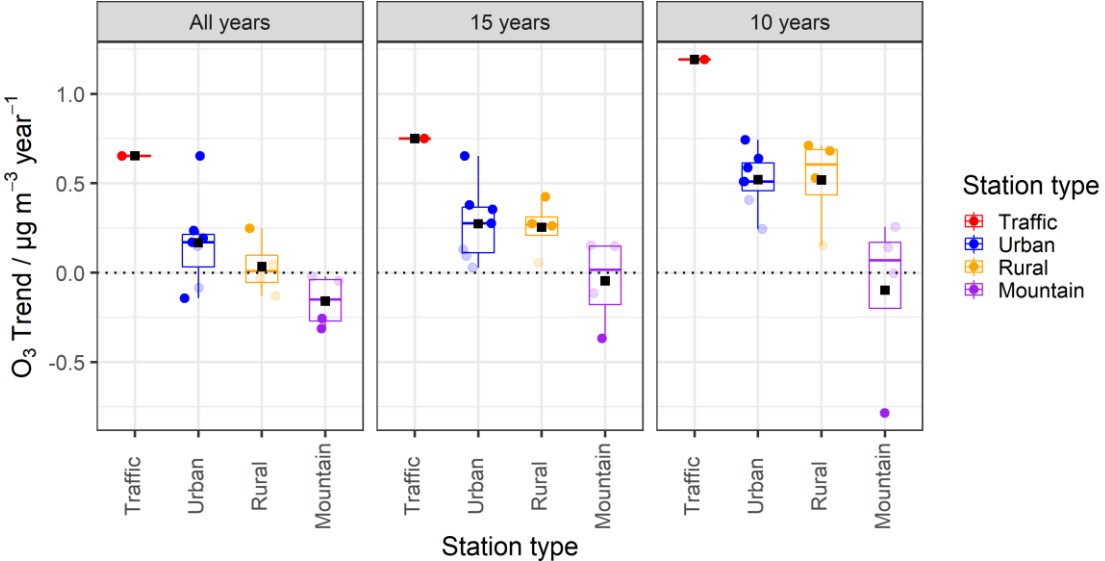

**Figure 3: Boxplots of observed O₃ trends over different time periods at the 4 station types. Transparent dots indicate statistically non-significant values. Black squares indicate means across all station values.**

*Traffic-dominated sites*

In Fig. 3 and Table S6, the increasing trend at the traffic site DD-Nord in the recent 10 years is roughly 1.2 µg m$^{-3}$ year$^{-1}$ or 3.50 % year$^{-1}$ and has nearly doubled as compared to 0.65 µg m$^{-3}$ year$^{-1}$ or 2.30 % year$^{-1}$ over the whole period since 1997. Similarly, most urban sites have also shown more rapid O₃ increase in the last decade, up to 0.74 µg m$^{-3}$ year$^{-1}$ in DD-Winkelmannstr., corresponding to about 1.75 % year$^{-1}$ and ranging from 0.24 to 0.64 µg m$^{-3}$ year$^{-1}$ (or 0.51 to 1.42 % year$^{-1}$) at the other urban background stations, although the trends in Plauen-DWD and Zittau-Ost are not statistically significant.

*Rural background sites*

In the rural background, with the exception of Niesky, an O₃ increase from 0.53 to 0.71 µg m$^{-3}$ year$^{-1}$ (or 0.92 to 1.49 % year$^{-1}$) is also found over 10 years, while over 15 years the trend is somewhat lower with 0.3 µg m$^{-3}$ year$^{-1}$ or 0.6 % year$^{-1}$ and during the entire period considered the trends at all stations, except Schkeuditz, are stagnant.

*Mountain sites*



At the four stations of the mountain ridge, the findings are least consistent. While at Fichtelberg a decreasing $O_3$ trend is consistently derived over all observation periods, amounting to about -0.8 µg m$^{-3}$ year$^{-1}$ (or -1 % year$^{-1}$) over 10 years and about -0.35 µg m$^{-3}$ year$^{-1}$ (or -0.4 % year$^{-1}$) over 15 and more years, the trends at the stations Schwartenberg and Zinnwald are

slightly positive over 10 and 15 years and slightly negative over the total period, but not statistically significant in all available periods considered. In Carlsfeld, trends are significant only for the longest observation period of ~20 years and with about -0.26 µg m$^{-3}$ year$^{-1}$ or -0.35 % year$^{-1}$ similarly high as at Fichtelberg, while over shorter periods they stagnate, in contrast to Fichtelberg. The reasons for the different behaviour of ozone at Fichtelberg are unclear, it might be related to its altitude of about 1200 m, which is the highest among the mountain stations, and a corresponding higher impact from ozone trends in

lower-stratosphere and free troposphere (Oltmans et al., 2013; Trickl et al., 2023).

*Ozone concentration decrements*

The trends identified for the four station types above, suggest that the typical concentration differences between station types also change over time. In order to consider this, "decrements", i.e. annual concentration differences between the different station types, were calculated and are shown as a time series in Fig. 4.

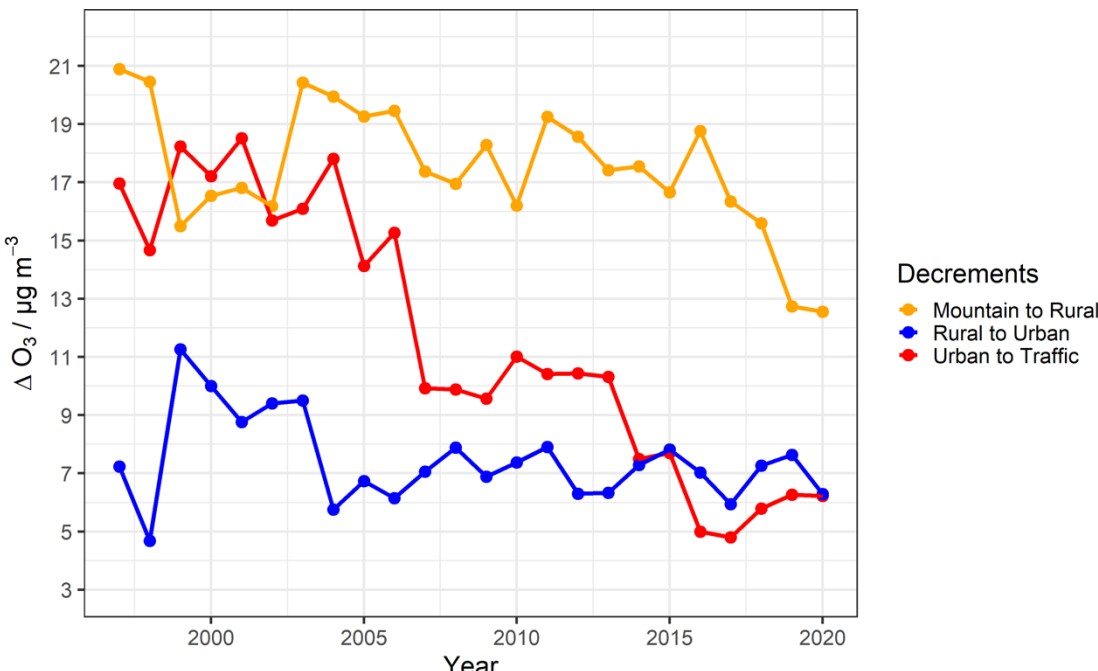

**Figure 4: Mean yearly decrements (concentration decreases) between different station types.**

It can be seen that the rural decrement, i.e. the concentration difference between the ore mountain stations and the rural

background, decreased from about 15 - 21 µg m$^{-3}$ at the turn of the century to values around 12 µg m$^{-3}$ in more recent years.





The urban decrement, i.e. rural to urban background difference, also decreased from around 10 µg m$^{-3}$ to about 5 µg m$^{-3}$. The traffic decrement from the urban background to the one long-term $O_3$ traffic station DD-Nord also decreased from about 15 to about 5 - 7 µg m$^{-3}$ and is now at a similar level as the urban decrement. If the $O_3$ trends mentioned above continue in a similar way in the future, it can be expected that the typical $O_3$ concentrations at the different station types will continue to converge, i.e. concentrations at traffic stations will become more similar to those in the urban background, these in turn will become increasingly similar to those in the rural background, and the latter might slowly approach the typical level at higher altitudes - albeit still with a clear gap. Due to the typical settlement densities, this would mean higher $O_3$ exposures for the vast majority of the population. The Air Quality Expert Group (2021) also reported for the UK that the urban $O_3$ concentrations have become gradually closer to rural areas from 2000 to 2019.

One recent study by Yan et al. (2018) using a statistical trend model based on the data from 685 sites from the European Environment Agency Airbase system and 93 European rural background sites monitored by the Chemical Coordination Centre of the European Monitoring and Evaluation Programme (EMEP) network, has reported statistically significant growth rates of annual mean $O_3$ (0.20 - 0.59 µg m$^{-3}$ year$^{-1}$) in suburban and urban sites for the period 1995 - 2014, in contrast to a slight $O_3$ decrease from -0.09 to -0.02 µg m$^{-3}$ year$^{-1}$ (without clear statistical significance) over rural background sites. Despite the time period not being directly comparable, this is roughly similar to the trends observed in Saxony between 1997 and 2020, which averaged around 0.2 µg m$^{-3}$ year$^{-1}$ for urban background stations and close to zero for rural background stations. Further, Finch and Palmer (2020) reported trends in annual mean $O_3$ in UK over the period 1999 - 2019 were also rather stagnant at rural background stations, with a mean increase of 0.16 µg m$^{-3}$ year$^{-1}$ (not statistically significant), and slightly higher at urban background sites, with a mean increase of 0.47 µg m$^{-3}$ year$^{-1}$ (statistically significant), than the mean increase of 0.17 µg m$^{-3}$ year$^{-1}$ observed in the comparable 1997 - 2020 time period in the Saxony urban background. Other analyses of the UK $O_3$ trend over the period 2000 to 2019 (The Air Quality Expert Group, 2021) suggested that only moderate increases in annual mean $O_3$ were observed at rural background sites, with a mean of 0.11 µg m$^{-3}$ year$^{-1}$, which is roughly between the mean values in Saxony (with quasi-zero and 0.25 µg m$^{-3}$ year$^{-1}$ for the periods from 1997 and from 2006 to 2020, respectively). In UK suburbs and urban areas, $O_3$ showed upwards trends, sometimes significantly, over the period considered in the study, with an average increase of about 0.26 µg m$^{-3}$ year$^{-1}$, which is also similar to the mean values at urban background stations in Saxony (0.17 and 0.27 µg m$^{-3}$ year$^{-1}$ over the last 24 and 15 years, respectively).

Despite all heterogeneity in detail, these comparisons suggest $O_3$ increases especially in urban areas and, to a certain extent, also in the rural background not only in Saxony, but in many places in Germany and Europe. The determined increases in the range of a few tenths of µg m$^{-3}$ year$^{-1}$ are not very high compared to the typical $O_3$ concentrations of approx. 40 - 60 µg m$^{-3}$ for urban and rural sites (as shown in Fig. 2), but they document $O_3$ still being a problem, at least with regard to chronic exposure, which has not been solved despite the successful reduction of various precursor compounds and even tends to increase in the future. This conclusion is also supported by the stronger $O_3$ increases in more recent years identified in this study and by Sicard (2021) as compared to the longer time periods often considered in other studies.





### 3.1.3 Trends for different O₃ concentration levels

In addition to the trends of mean $O_3$ concentrations considered so far, it is interesting to also investigate concentration changes depending on the absolute $O_3$ concentrations. This can be achieved by calculating the trends of different $O_3$ percentiles. Low percentiles, i.e. 0, 1st, 5th and 10th, indicate the lowest and low concentration levels, medium percentiles, i.e. 25th, 50th and 75th, indicate middle concentration ranges, and high percentiles, i.e. 90th, 95th, 99th and 100th, indicate trends at high and highest $O_3$ levels.

These trends over four station types within the three periods defined above are shown in Fig. 5.

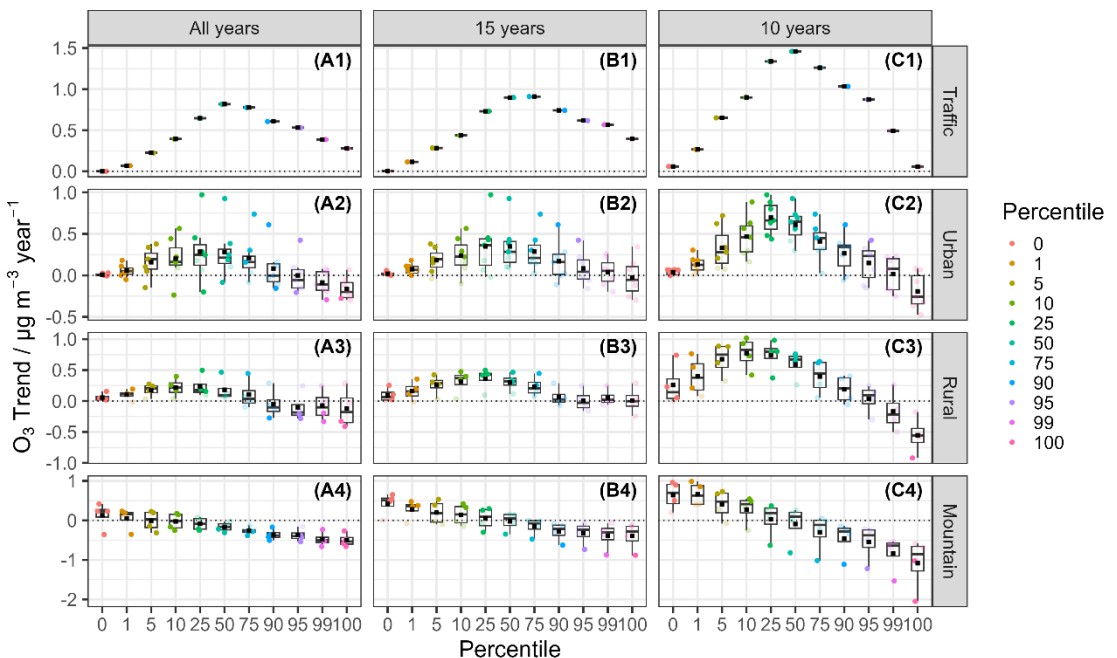

**Figure 5: Theil-Sen trend values for different percentiles of O₃ concentrations in four station types and for three time periods until 2020, respectively. Coloured dots show the values of individual stations, with solid dots for statistically significant trends and transparent dots otherwise. All dots are summarized by the boxplots. The black square reflects the mean trend value per percentile.**

*Traffic site*

The traffic-influenced site shows ozone increases for all concentrations (0.06 - 1.46 µg m⁻³ year⁻¹) and the strongest trends at median concentrations around the 50th percentile combined with nearly no increases at the very low and very high percentiles.

This behaviour leaded to the depicted bell-shaped curves. The longer the considered time frame, the smaller is the dynamic range of the ozone trends from median against more extreme concentration regimes. During the recent decade, the traffic site has exhibited the most pronounced increases across the entire concentration range.



*Urban and rural background sites*

For the urban and rural background there are quite similarly increasing trends (0.01 - 0.97 µg m$^{-3}$ year$^{-1}$) for nearly all
concentration levels with the important exception that very high O$_3$ concentrations actually decrease (-0.54 - -0.06 µg m$^{-3}$ year$^{-1}$) and the strongest increases were seen already at the 10$^{th}$ and 25$^{th}$ percentiles, respectively and not only at the 50$^{th}$. Each O$_3$
concentration level has also shown a greater increase over the past decade or a more pronounced decrease at the highest
concentrations compared to the other two periods. Additionally, the range of differences between peak values in the, again,
bell-shaped curves during the recent decade, is dampened against the traffic site and amounts to about 1 µg m$^{-3}$ year$^{-1}$.

*Mountain sites*

For all the mountain sites a monotonous decrease in the O$_3$ is seen with increasing concentration percentiles and only for the
smallest percentiles O$_3$ increasing trends are seen which switch to O$_3$ decreasing trends at the 5$^{th}$, 25$^{th}$ and 50$^{th}$ percentiles for
the all years, 15 years and 10 years, respectively. There is no bell-shape anymore but a generally linear change.

Quantitatively, there are increases (0.05 - 0.66 µg m$^{-3}$ year$^{-1}$) in percentiles from the minimum to the 10$^{th}$ turning into decreases
(-1.08 - -0.28 µg m$^{-3}$ year$^{-1}$) in the higher percentiles (90$^{th}$ - 100$^{th}$). The shorter the time frame considered, the more pronounced
is the stated trend.

Overall, during recent 10 years, stagnant or even downward trends in the O$_3$ 90$^{th}$, 95$^{th}$, 99$^{th}$, and maximum of O$_3$ in urban, rural
and mountain stations had occurred, while at the lower percentiles (minimum, 1$^{st}$, 5$^{th}$ and 10$^{th}$), O$_3$ concentrations across all
stations seemingly continued to increase (with a mean trend ranging from 0.03 - 1.0 µg m$^{-3}$ year$^{-1}$). In the range of medium
percentiles (from 25$^{th}$ to 75$^{th}$), all urban and rural sites showed statistically significant increasing trends (ranging from 0.4 to
1.46 µg m$^{-3}$ year$^{-1}$), whilst at the mountain sites, they were decreasing or stagnant (-0.3 - 0.03 µg m$^{-3}$ year$^{-1}$).

Quite recently, Finch and Palmer (2020) similarly reported no statistically discernible decreasing trend (-0.49 µg m$^{-3}$ year$^{-1}$) in
annual maximum O$_3$ on average, while mean concentrations and minimum concentrations increased with 0.41 and 0.09 µg m$^{-3}$ year$^{-1}$, respectively, across rural, suburban and urban background sites in the UK over the period 1999 - 2019. These findings
can, at best, be compared to panels A2 and A3 of Fig. 5 in illustrating O$_3$ trends in the similar period from 1997 - 2020, with -0.03 µg m$^{-3}$ year$^{-1}$ in maximum O$_3$, 0.23 µg m$^{-3}$ year$^{-1}$ in median concentration and 0.02 µg m$^{-3}$ year$^{-1}$ at minimum concentration.
The results of another study in the UK (The Air Quality Expert Group, 2021) also revealed that the 50$^{th}$ and 25$^{th}$ percentiles
concentrations of O$_3$ have clearly increased (from 0 to 0.94 µg m$^{-3}$ year$^{-1}$) at urban background sites over the period 2000 -
2019 (cf. panel A2 of Fig. 5 with 0.27 - 0.31 µg m$^{-3}$ year$^{-1}$ in the similar period from 1997 - 2020), whilst corresponding
increases (from 0 to 0.66) at rural sites (cf. panel A3 of Fig. 5 with 0.07 - 0.30 µg m$^{-3}$ year$^{-1}$) are in general smaller (and in
many cases not statistically significant). In addition, there have been stagnant or significant downward trends (-2.05 to 0.64
µg m$^{-3}$ year$^{-1}$, despite reported positive values are non-statistically significant) in the upper percentiles (99$^{th}$ and 99.9$^{th}$
percentiles) across all 27 sites examined, similar with our result of -0.26 - 0.13 µg m$^{-3}$ year$^{-1}$ in 99$^{th}$ and 100$^{th}$ (cf. panel A2
and A3 of Fig. 5).



Our results are consistent with previous analyses with negative or stagnant trends in the high range of percentiles ($99^{th}$ - $100^{th}$), indicative of fewer extreme $O_3$ episodes over time, which might be related to the reductions in $NO_x$, VOCs and CO emissions (Derwent et al., 2010; Yan et al., 2018 and references therein). The trends in lower and middle percentiles of $O_3$ are almost always positive, corresponding to the increasing baseline level of $O_3$ in the northern hemisphere (Jonson et al., 2006; Jenkin, 2008; Dentener et al., 2010; Cooper et al., 2014). Yan et al. (2018) used sensitivity simulations and statistical analysis to report

that a decrease in European anthropogenic emissions had lowered the $95^{th}$ percentile of $O_3$ concentrations but enhanced the $5^{th}$ percentile of $O_3$ in rural, suburban and urban sites during 1995 - 2014 over Europe. The results described here (decreasing trends in higher range of percentile ($95^{th}$ - $100^{th}$) of $O_3$ and increasing trends in lower range of percentile (0 - $5^{th}$) of $O_3$ as shown Fig. 5 A2 - A4) also may reflect the effectiveness of emissions reductions in controlling highest-level ozone for Saxony but also show an opposite effect at low concentration levels.

**3.1.4 Trends at annual seasons**

The mean trends by season at the different station types within the three time spans as used before are summarised in Fig. 6. For the traffic site, increasing trends are observed in all seasons over all time periods, but strongest in summer or spring. In urban and rural background areas, a similar seasonal pattern was found for the recent 10 years, but for longer periods, spring or summer show lower increasing trends than autumn or winter. Particularly for mountain sites in spring and autumn, the

trends during the longer periods are negative in contrast to the at least partly positive trends for the most recent decade. Yan et al. (2018) reported increasing trends ($\sim 0.10$ µg m$^{-3}$ year$^{-1}$) in winter mean $O_3$ and decreasing trends ($\sim -0.30$ µg m$^{-3}$ year$^{-1}$) in summer mean $O_3$ for the period 1995 - 2014 in suburban, urban and rural sites in Europe, which is similar to the trends observed here for all years (1997 - 2020), cf. Fig. 6 panels A2 and A3 with around 0.4 and 0.1 µg m$^{-3}$ year$^{-1}$ in winter and summer, respectively.






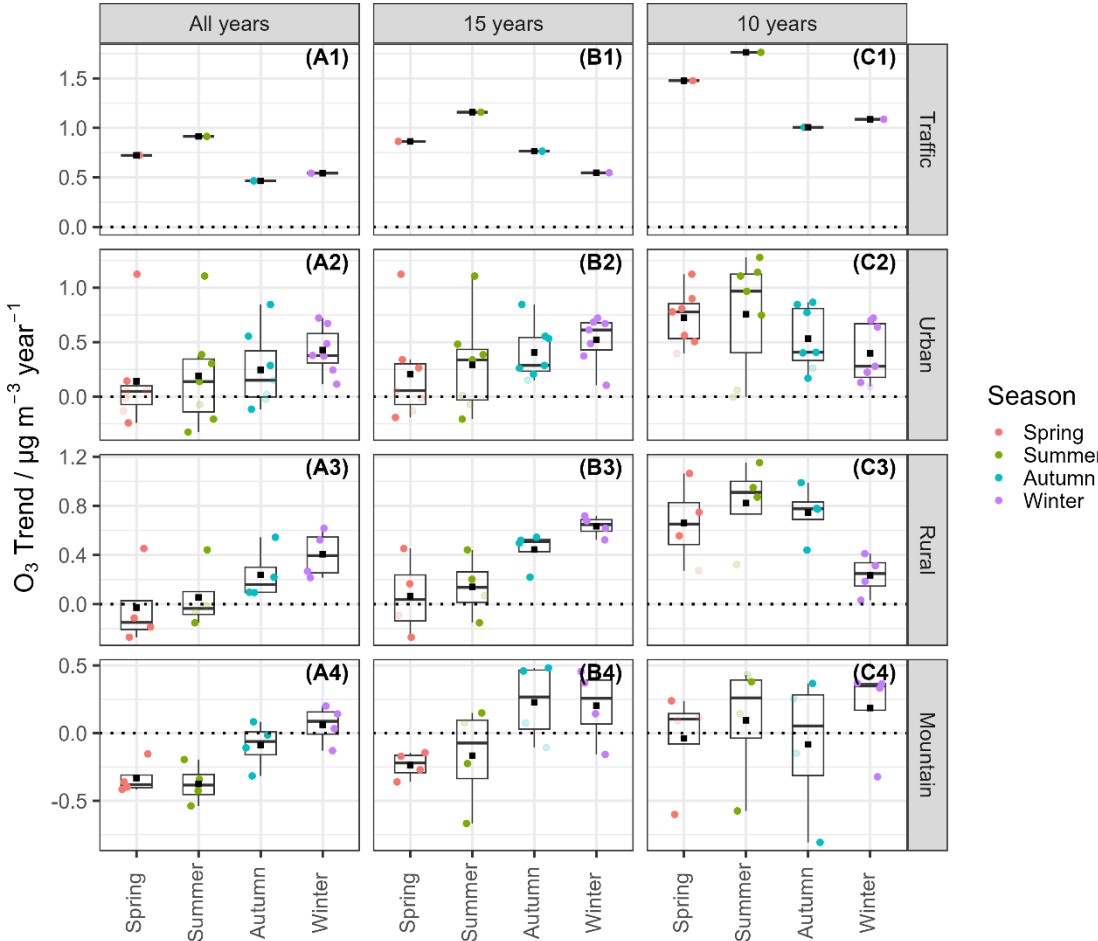

**Figure 6: Theil-Sen trend values at different times of the year at the various station types and within each of the three time spans up to 2020. Coloured dots show the values of individual stations, each of which is summarised using the boxplots. The black square shows the mean trend value per season.**


Gebhardt et al. (2021) also analysed a data set of O₃ concentrations on the ore mountain ridge (Carlsfeld, Fichtelberg, Schwartenberg and Zinnwald) and found stagnating trends in the winters from 1997/98 to 2020, which is consistent with the result of mountain stations in this present study, see Fig. 6 panel A4.

**3.2 Relationships between trends in O₃ and other measured trace gases concentrations and other parameters**

In order to examine how the observed O₃ trends relate to the trends of other air pollutants, the trends for nitrogen oxides and "oxidant" (Oₓ) as the sum of O₃ and NO₂ (Kley et al., 1994) are shown in Table S7 for the three time periods. Nitrogen oxides (NO, NO₂ and NOₓ) show significantly decreasing trends since 1997 and over the last 10 years, consistent with reports that Europe-wide emissions of O₃ precursors (NOₓ and other air pollutants) have decreased substantially since 1990 (Colette et al.,



2015b; The Air Quality Expert Group, 2021). The MACCity inventory shows that anthropogenic $NO_x$ emissions in Europe
decreased by 35% between 1995 and 2015 (Yan et al., 2018). This is comparable to the results identified here, which
correspond to an average decrease of 31% at all sites over the last decade (2011-2020). In addition, the inventories reported
by the individual European countries can be found in the "Air Pollutant Emissions Data Viewer"
(https://www.eea.europa.eu/data-and-maps/dashboards/air-pollutant-emissions-data-viewer-3) of the European Environment
Agency (EEA). This database shows that $NO_x$ emissions in Germany have decreased by about 31 % from 2011 to 2020, which
is again consistent with the mean trends of nitrogen oxides reported here.

In addition to the directly measured air pollutants, it is useful to consider the trend of the so-called "oxidant" ($O_x$), which results
from the sum of $O_3$ and $NO_2$ (Kley et al., 1994). In this way, the long-term change of the two oxidants can be better understood
and evaluated, taking into account the effect of NO titration ($NO + O_3 = O_2 + NO_2$) in the near-surface atmosphere.

Most stations in Saxony show stagnating or increasing $O_3$ (see Sect. 3.1), but Table S7D shows slightly decreasing $O_x$ trends
for all stations, which are, however, with the exception of the traffic station, only statistically significant for the longer period
($> = 15$ years). The traffic station shows decreasing $O_x$ trends with -0.26 µg m$^{-3}$ year$^{-1}$ during the last 10 years, which means
that $NO_2$ has decreased slightly more than $O_3$ has increased. The $O_x$ trend has tended to be stagnant (non-significant) over the
10 years at urban (-0.1 µg m$^{-3}$ year$^{-1}$), rural (0.0 µg m$^{-3}$ year$^{-1}$) and mountain sites (-0.12 µg m$^{-3}$ year$^{-1}$), while similar decreasing
trends as at the traffic station are observed for the longer period ($\sim$ -0.2 µg m$^{-3}$ year$^{-1}$). At a busy road station in London, a local
decrease in $O_x$ of about 38% over the period 2011 - 2020 was reported (The Air Quality Expert Group, 2021), which is
significantly more than the decrease in traffic site (DD-Nord) with about 0.68 % in the same period.

Overall, the $O_x$ observations show that the increase in $O_3$ observed at some stations in Saxony is probably related to the
decreasing $NO_x$. However, the stagnating $O_x$ trends, especially in the more recent 10 years, show that the achieved emission
reductions of the $O_3$ precursors obviously did not lead to sustainably and significantly lower $O_x$ concentrations and thus to
lower exposure to $NO_2$ and $O_3$ on average.

To understand the relationship between $O_3$ precursors and $O_3$ trends, the close chemical relationship between $O_3$ and the
nitrogen oxides can be first investigated in Fig.s 7 and S3 where the trends of $NO_x$, NO and $NO_2$ at all stations and for the
different time periods are plotted above the respective trends of $O_3$ at the sites.



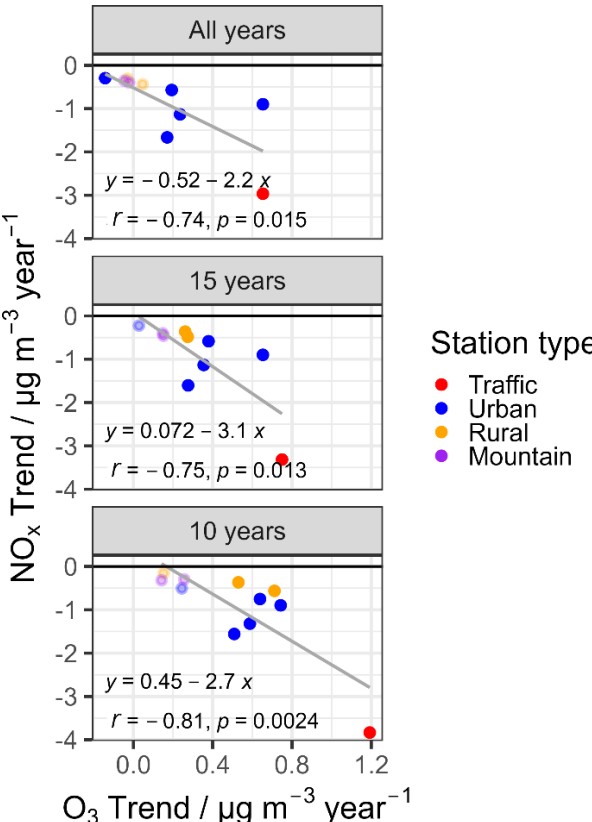


**Figure 7: Relationships between O₃ trends and NOₓ trends across all stations for the 3 different time periods. Transparent dots indicate statistically non-significant O₃ trends.**

Regardless of the time period considered, it can be seen that the more the O$_3$ concentration at a station has been increasing, the more the nitrogen oxide concentrations have generally been decreasing. This pattern applies to all stations and is possibly a direct result of the classical Leighton chemistry, which describes that, during the day, a reduction in nitrogen oxide concentrations leads to a shift in the photostationary equilibrium towards higher O$_3$ concentrations. At night, lower NO$_x$ concentrations mean a reduction in the sink strength of O$_3$ via the reaction with NO$_2$ and thus also higher resulting O$_3$ concentrations. In sum, the opposing trends of nitrogen oxides and O$_3$ trends result in the only very slightly decreasing and often stagnating trends of the entire O$_x$ described above.

## 3.3 Photochemical model results

In Sect. 3.2 it has been found that there is a consistent increase in O$_3$ levels at monitoring stations, coupled with a noticeable decline in NO$_x$ concentrations, particularly evident in many urban regions. Moreover, Fig.s S4 and S5 show how the reduction in NO$_x$ concentrations has correspondingly led to elevated O$_3$ levels across the four station types, both in summer and winter seasons spanning the period from 2000 to 2019. In this section, seasonal isopleths of O$_3$ formation will be diagnosed to offer





insights into the controlling effects of $NO_x$ and VOCs on $O_3$ variations across different station types throughout the 2000 - 2019 timeframe. By visualizing the station type-to-type variations within two seasonal isopleths (see the Sect. 2.3 for detailed information), the present model-based analysis seeks to gauge the efficacy of precursor controls in Saxony over the past two decades, and help shed some light on future mitigation strategies targeting $O_3$ pollution.

### 3.3.1 Simulation setup and comparison with 2019 measurements


As a first step in model development, the performance of the base cases simulations for Saxony in 2019 was assessed. The year 2019 was chosen because of the best availability of the required input parameters. As a first set of results, the modelled NO and $NO_2$ diurnal patterns during summer and winter are compared with observed data at rural background stations in Fig. 8. It generally shows a good agreement between the modelled and observed NO and $NO_2$ diurnal patterns, particularly during

the day. At night, the model typically predicts concentrations of less than 1.0 µg m$^{-3}$. However, the measured diurnal patterns of the NO are influenced by the detection limit (DL) of the NO monitor (DL=1.0 µg m$^{-3}$) and any values below the DL are given as the DL. So, in Fig. 8 the modelled values are plotted with an increase of 1 µg m$^{-3}$ in order to provide a better comparison with the measured NO. Accordingly, the comparison of the nocturnal NO patterns is only possible for hours with measured NO above the DL. The modelled NO concentrations during the period from 06:00 to 11:00 (CET) demonstrate satisfactory

agreement with the observed summer data (Fig. 8A), with the exception of the Niesky site. During winter (Fig. 8B), the model seems to underpredict the NO concentrations maybe because of a too high mixing layer being considered. However, the temporal pattern agrees rather well. Therefore, the linear correlation coefficient (r) exceeds 0.8 in summer and remains above 0.7 in winter, indicating a robust correspondence between model outputs and measurements for both seasons.





**Figure 8: Diurnal profiles of hourly averaged modelled and observed NO and NO₂ in Saxony rural background (A, C for summer case, and B, D for winter cases). Shaded areas indicate night-time, black line indicates the modelled values, and coloured lines refer to the observed concentrations. Noted that all observed NO concentrations are all greater than 1 µg m⁻³ (DL), that means values below 1 µg m⁻³ default to 1 µg m⁻³. So the modelled values are plotted with an increase of 1 µg m⁻³ in order to provide a better comparison with the measured NO.**

For NO$_2$ it can be seen that in summer (in Fig. 8C) the simulated NO$_2$ exhibits a one-hour earlier peak at 5:00 in the morning and a one-hour delay in the afternoon at 15:00. For all sites r between modelled and measured averaged NO$_2$ is at least 0.7. During the winter months, there is no correlation for the entire 24h period, but it should be noted that at all sites is above 0.7 during daytime hours, specifically between 08:00 and 15:00 (in Fig. 8D). Overall, while the simulation captures the diurnal variations of NO and NO$_2$ reasonably well, a notable discrepancy persists partly between the simulated and observed





concentrations levels of $NO_x$. This inconsistency likely stems from the inherent uncertainties within the local emission inventory, particularly regarding the accurate estimation of local-site $NO_x$ emission rates, or overestimation of the local mixing layer height also influencing the pollutant emissions rates.

Figure 9 shows the diurnal profiles of hourly averaged simulated and observed $O_3$ concentrations during summer and winter. The model effectively reproduces both the magnitudes and diurnal patterns of observed $O_3$, demonstrating a strong correlation between observation and simulation (r > 0.8). However, noteworthy exceptions include a higher simulated $O_3$ peak occurring after 13:00 and a one-hour delay in its occurrence. These deviations can be attributed to very localized near-site $NO_x$ emissions (as depicted in Fig. 8), which cannot be adequately captured by the simulations.


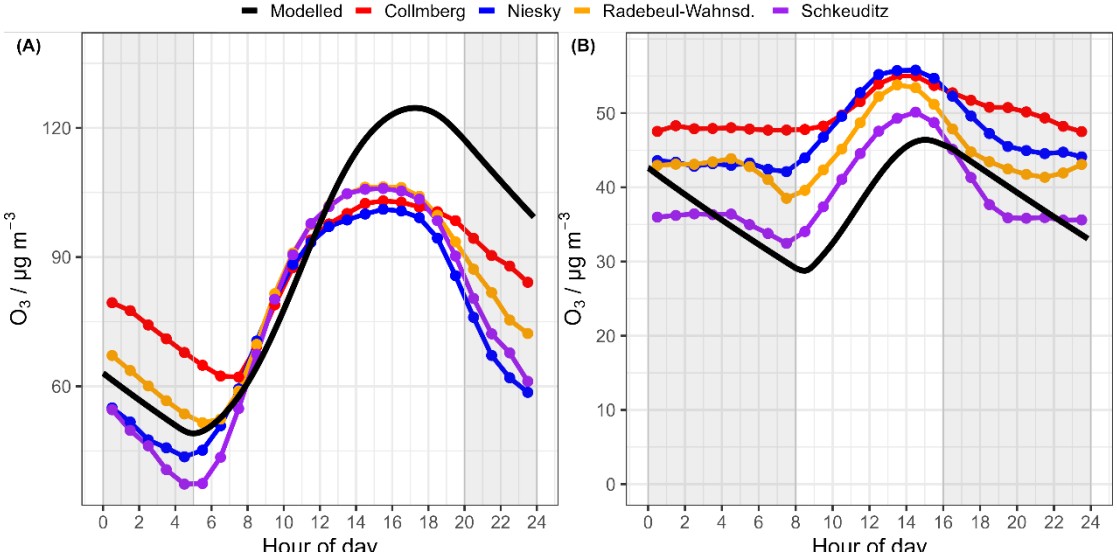

**Figure 9: Comparison of the hourly averaged modelling results with the observed $O_3$ concentrations in Saxony rural background (A and B for summer and winter cases, respectively). Shaded areas indicate night-time, black line indicates the modelled $O_3$, and coloured lines refer to the observed $O_3$.**


Overall, Fig.s 8 and 9 reveal a reasonable agreement between the observed and simulated datasets. Therefore, the simulations were found to be generally able to reproduce the $O_3$ photochemistry for the conditions present in Saxony in 2019, enabling further sensitivity simulations on the $O_3$ and its characteristic dependencies on both the $NO_x$ and VOCs conditions.

### 3.3.2 Isopleths for $O_3$ formation

Both Sect. 3.1 and 3.2 have highlighted a significant increase in $O_3$ levels in recent years. It is essential to elucidate the relationships between $O_3$ and its precursors in order to develop science-based control measures. Therefore, hundreds of sensitivity simulations were performed, incorporating various emission rates multiplier (outlined in Table S5) of TNMVOC and $NO_x$. These simulations aimed to elucidate the $O_3$ production rate with regards to the ambient concentrations of $NO_x$ and



TNMVOC, which is depicted in the ozone isopleths of Fig. 10. A brief description how these are produced is provided in

experimental Sect. 2.3. It can be seen that the diagrams resemble classic ozone isopleths typically produced with models

(Sillman et al., 1990; Ehlers et al., 2016). They depict the in-situ NetPO₃ as a function of NO$_x$ and TNMVOC concentrations

for both summer and winter scenarios.

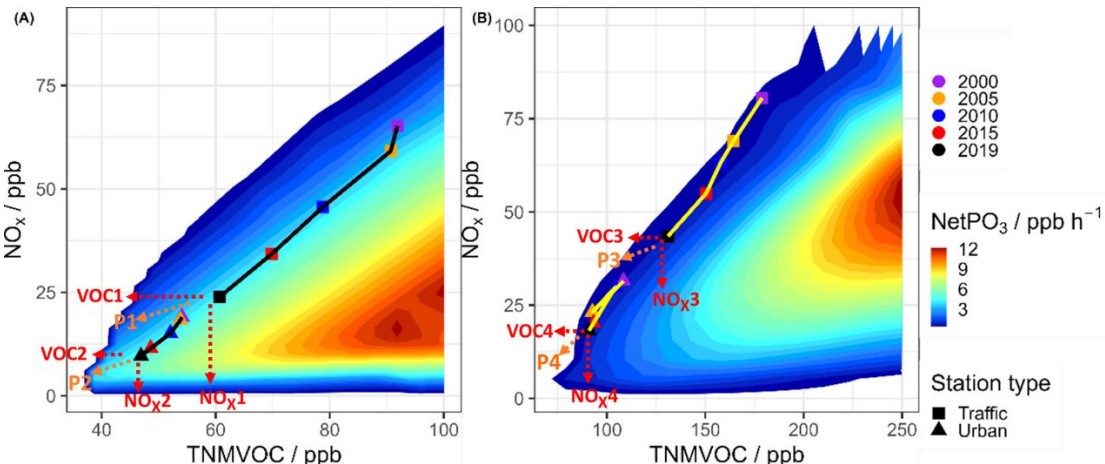

**Figure 10: Isopleth plots for net O₃ production rate (NetPO₃ in ppb h⁻¹) during 12:00 - 13:00 CET as a function of averaged NO$_x$ and TNMVOC concentration (A and B for summer and winter cases, respectively). Red dotted arrow lines indicate a hypothetical reduction in only the one precursor (TNMVOC (VOC1 - VOC4) or only NO$_x$ (NO$_x$1 - NO$_x$4)) in the future. Pale orange dotted arrow lines suggest potential future pathways for further reduction in TNMVOC as well as moderate NO$_x$ reduction (P1 - P4).**

In the next step, the measured values of NO$_x$ and observed O₃ change rate (dO₃/dt) for the years 2000, 2005, 2010, 2015, and

2019 were used to indicate for each year the location in the isopleth diagram. A key challenge, however, was the lack of

measured TNMVOC concentrations in all years. To work around this, they were estimated based on measured NO$_x$ and

measured O₃ change rate (dO₃/dt) from 06:00 to 12:00 and 08:00 to 12:00 for summer and winter, respectively. As depicted in

Fig. S6, measured dO₃/dt and modelled NetPO₃ agreed reasonably well during these times of the day, which is why it is

considered valid to interchange them in the present application. The grid of modelled NetPO₃ as a function of modelled,

inventory-derived NO$_x$ and TNMVOC concentrations (see Sect. 2.3), was therefore interpolated to derive TNMVOC

concentrations for given measured NO$_x$ and dO₃/dt. For further clarification, in Tables S8 and S9, these estimates are shown

together with a comparison of measured and modelled NO$_x$ and dO₃/dt.

Notably, simulated TNMVOC concentrations, primarily emitted or locally photochemical in origin, are expected to be similar

to or lower than previous measurements (Knobloch et al., 1997) because of lowered emissions through environmental

mitigations. Indeed, unpublished data for a range of NMVOCs observed throughout the year 2022 in Borna, a city south of

Leipzig, exhibited remarkably low concentrations of 66 NMVOCs species (Table S10) at a near-road measurement site, with

hourly maximum total mixing ratios of only 29.7 ppb in summer and 204.6 ppb in winter (Table S11). Besides, von





Schneidemesser et al. (2018) reported that the highest mixing ratios of total 57 NMVOCs compounds measured during 3

summer months (June-August) at traffic sites of central and western Berlin were found with 64 ppb and 170ppb, respectively. Thus, our estimated summer and winter TNMVOC concentrations (Tables S8 and S9) can be considered to lie in a reasonable range. However, for rural and mountain sites, accurate estimations for the TNMVOC were challenging due to their very low $NO_x$ concentrations (< 10 ppb in summer, < 16 ppb in winter) and $dO_3/dt$ (< 4 ppb $h^{-1}$ in summer, < 1.3 ppb $h^{-1}$ in winter) observations (refer to Tables S8 and S9). Especially in the last 10 years, the data are near the lower-left border of simulated

$NO_x$ and $NetPO_3$ values in Fig. 10, where there exists considerable uncertainty in estimating $NetPO_3$ and TNMVOC. Therefore, only data for traffic and urban background sites are shown in Fig. 10.

The modelled $NetPO_3$ determined in the present study were in the range from 2.74 to 4.87 ppb $h^{-1}$ in summer (Table S8) and 0.78 to 1.65 ppb $h^{-1}$ in winter (Table S9) at traffic and urban sites between 2000 and 2019. In rural and mountain sites, the $NetPO_3$ is not given for the different years due to the uncertainty of the estimates, as described above. However, it can be said

that measured $dO_3/dt$ ranged from 1.16 to 3.89 ppb $h^{-1}$ in summer and 0.35 to 1.30 ppb $h^{-1}$ in winter from 2000 to 2019. Either modelled $NetPO_3$ or measured $dO_3/dt$ were comparable to those deduced from other sites in Germany and Europe. For instance, Corsmeier et al. (2002) reported a $NetPO_3$ of 4.0 - 6.5 ppb $h^{-1}$ by a model system (KAMM/DRAIS) for a polluted ozone plume from Berlin during the July 1998 BERLIOZ campaign. In relatively clean rural and mountain sites, during the FREETEX'96 study at Mt. Jungfraujoch in April/May 1996, the $NetPO_3$ were calculated using a photochemical box model, yielding rates of

0.27 and 0.13 ppb $h^{-1}$ (Zanis et al., 2000), and Nussbaumer et al. (2021) reported a maximum midday $NetPO_3$ of 1.5 ppb $h^{-1}$ based on a photochemical calculation, considering in situ trace gas observations at a boreal forest site in Finland during July and August 2010 HUMPPA campaign.

As can be seen from Fig. 10, $O_3$ photochemical formation in densely populated regions of Saxony is currently VOC-controlled and has also been VOC-controlled since the year 2000, either in summer or in winter. In detail, for the summer case in Fig.

10A, traffic and urban background stations have a clear temporal trend, and were in the VOC-limited regime overall from 2000 to 2019, but for the urban background stations the trend line in the recent 10 years is close to the transition regime (from VOC-limited to $NO_x$-limited regime). Detailed analyses (refer to Fig. 10A and Table S8) reveal a different trendline pattern in $NetPO_3$ between traffic and urban stations. At the traffic station, beginning at 2.74 ppb $h^{-1}$ in 2000, the $NetPO_3$ trendline crossed several isopleths, reaching 3.69 ppb $h^{-1}$ by 2005. Subsequently, it maintained a nearly parallel course with isopleths, starting

with 3.87 ppb $h^{-1}$ in 2010 and then rising to 4.55 ppb $h^{-1}$ in 2019 over the recent decade. Concurrently, both $NO_x$ and TNMVOC concentrations decreased by two-thirds and one-thirds, respectively. Conversely, urban background sites depict a slightly different scenario. An increase in $NetPO_3$, from 4.13 ppb $h^{-1}$ in 2000 to 4.87 ppb $h^{-1}$ in 2019, is noted within the isopleth plot. This rise occurred alongside substantial reductions in $NO_x$ concentrations by 50% (from 19.02 to 9.61 ppb), contrasting with only a small decrease of TNMVOC by approx. 15% (54.06 to 46.91 ppb). Given that both traffic and urban sites are

characterised by a VOC-limited regime, where $NetPO_3$ increases with decreasing ratios of $NO_x$/TNMVOC, the observed increase in $NetPO_3$ and $O_3$ levels (as depicted in Fig. S7) confirms the inadequacy of current efforts of TNMVOC emission



reductions with respect to $O_3$ pollution. Stronger reduction measures over the next years are needed, particularly in areas with dense population concentrations, to avoid exacerbating rather than mitigating ozone pollution.

In contrast to summer, winter experiences substantially weaker photochemistry, attributed to the lower intensity and shorter

duration of solar radiation, which is merely half of that in summer. As illustrated in Fig. 10B and Table S9, traffic and urban background stations have exhibited relatively low $NetPO_3$ (< 2.00 ppb h$^{-1}$) over the past two decades. Nevertheless, a slight increase in $NetPO_3$ has been observed (from approximately 0.80 in 2000 to about 1.60 ppb h$^{-1}$ in 2019 for traffic stations, and from about 1.40 to roughly 1.60 ppb h$^{-1}$ for urban background sites, respectively), despite halving $NO_x$ and TNMVOC emissions. Winter $O_3$ formation at both station types has predominantly occurred in the VOC-limited regime over the past 20

years, indicating that traffic and urban sites still have a considerable way to go in achieving $NO_x$-limited conditions despite significant $NO_x$ emission controls. Additionally, it's worth noting that trendlines tend to run more parallel in the recent 5 or 10 years, irrespectively of the season, suggesting that $O_3$ formation via photochemistry in densely populated regions under current emission controls has remained more consistent in recent years, contributing to $O_3$ increases (see Fig.s S7 and S8).

As described above, $NetPO_3$ trends at rural and mountain sites could not be discussed using the modelled isopleths. Instead,

trends in measured $dO_3/dt$ are briefly discussed in the following. From summer 2000 to 2019, there was an increase in $dO_3/dt$, by 1 ppb h$^{-1}$ and 0.5 ppb h$^{-1}$ (refer to Table S8) for rural and mountain sites respectively. The rise in summer $O_3$ levels (refer to Fig. S7) may be attributed to an increase in photochemical production due to more on-site or transported $O_3$ precursors or an increase in the hemispheric background $O_3$ level due to elevated anthropogenic emissions in heavily polluted areas (Derwent et al., 2015; Gaudel et al., 2018; Mertens et al., 2020). The positive trends observed in the lower and middle percentiles of $O_3$

across all sites (see Fig. 5) in the present study further reinforce the assertion that background $O_3$ is rising. During winter from 2000 to 2019, rural sites experienced a slight increase in $dO_3/dt$ by 0.2 ppb h$^{-1}$, while mountainous stations showed a decrease of -0.3 ppb h$^{-1}$ (refer to Table S9). Notably, winter $O_3$ trends increased at both sites, as depicted in Fig. S8, more likely due to the increase in the background $O_3$ level (Derwent et al., 2015; Gaudel et al., 2018; Mertens et al., 2020), as photochemistry does not change significantly during the winters of the past 20 years.

**3.3.3 Implications for $O_3$ control**

In the timeframe of the present study, anthropogenic emissions of NMVOCs and $NO_x$ have been significantly decreased in the whole of Saxony (Fig.s 11 and S9), as well as selected traffic (Fig.s S10 and S11) and urban background areas (Fig.s S12 and S13) over the last 20 years (from 2000 to 2019). Although the biogenic emissions data of year 2019 in overall Saxony (Fig. S14) are comparable to anthropogenic NMVOCs in same year (Fig. 11A), biogenic emissions of isoprene and alpha-pinene in

selected urban stations (Fig. S15) are indeed several orders of magnitude smaller than the anthropogenic emissions data in these areas (Fig. S12A). Limonene, on the other hand, cannot be regarded solely biogenic but has important anthropogenic sources as well (Borbon et al., 2023; Gu et al., 2024). Thus, in urban sites it is inferred that biogenic emissions have a negligible effect on $O_3$ formation. The seasonal $O_3$ isopleths suggest current $O_3$ formation regimes across traffic and urban background sites were determined to be overall VOC-limited during the same period. However, what kind of VOCs should be controlled




to reach the better O₃ decrease is still uncertain. Based on the inventory data available for the present study (Fig. 11), it can be noted that the solvent emissions decreased less strongly than the total anthropogenic NMVOCs emissions in Saxony, indicating an increased share of this emission category to regional O₃ formation. Similar changes in solvent emissions relative to total anthropogenic NMVOCs emissions were also observed at traffic (Fig. S10) and urban background stations (Fig. S12). In fact, the percentage of solvent emissions in total anthropogenic NMVOCs emissions remained nearly constant from 2010 to 2019,

regardless of station types. This consistency suggests that solvent emissions may be a potential contributing factor to the increase in O₃ levels over the past decade. A recent study in the UK suggested an increased importance of solvents as well for summertime urban O₃ formation by incorporating detailed VOCs emission inventories from 1990 to 2019 into a zero-dimensional chemical box model constrained by observational data (Li et al., 2024). Therefore, it is suggested that more strict VOCs emission controls be implemented in the future, possibly with more attention on solvent use, in order to alleviate the O₃

pollution in Saxony.

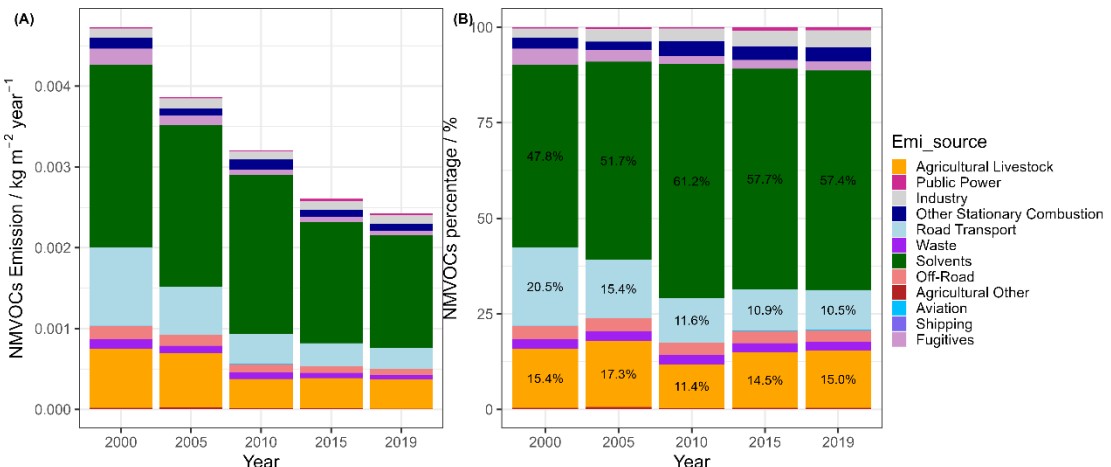

**Figure 11: Saxony anthropogenic emissions of NMVOCs, which were obtained from official UBA data. Emission data were averaged over the selected Saxony regions (Fig. S1).**


However, a scenario that only reduce VOCs (as indicated by arrows VOC1 - VOC4 in Fig. 10) is not realistic as it comes with the cost of constant NOₓ. Therefore, a scenario (as indicated by arrows P1 - P4 in Fig. 10) where NetPO₃ either slightly decreases, achieved through strong VOCs emission control with moderate NOₓ emission decrease, is more realistic. In view of the detailed effect from diverse NMVOCs emissions on O₃ pollution, more NMVOCs measurements should be implemented

to validate emission inventories and better refine modelling studies on O₃ formation.

**4 Conclusions**



Given the importance of surface ground-level $O_3$, long-term observations of $O_3$ concentrations are of utmost importance to understand the $O_3$ formation/depletion trend in a changing environment and climate. The present study compares the observed trends in $O_3$ concentrations across 16 measurement stations in Saxony, categorized into four station types (traffic, urban background, rural background, and mountain sites), over three distinct periods: i) the entire duration of available measurement data, from 1997 or later, ii) the 15-year span from 2006 to 2020, and iii) the more recent 10 years from 2011 to 2020. At 15 of the 16 measurement stations, surface $O_3$ exhibits upward or stagnant trends over the recent 15 or 10 years. The strongest $O_3$ trend is observed at the traffic DD-Nord station with roughly 1.2 µg m$^{-3}$ year$^{-1}$ (or 3.5 % year$^{-1}$) in the last 10 years. Increasing $O_3$ trends are also observed at urban and rural background stations, with average rates of about 0.5 µg m$^{-3}$ year$^{-1}$ (or 1.1 % year$^{-1}$) over the last decade. A more inhomogeneous picture can be seen on the mountain ridge, where over the last decade two stations (Schwartenberg and Zinnwald) show slightly positive but with not statistically significant $O_3$ trends, one station (Carlsfeld) was seen with a stagnating trend and at the highest station Fichtelberg the concentrations have clearly decreased with -0.79 µg m$^{-3}$ year$^{-1}$ (or -0.95 % year$^{-1}$). In addition, ground-level $NO_x$ measurements at these sites were analysed. Our results highlight that $O_3$ pollution in Saxony has not abated and has, in fact, worsened in the last 10 years, particularly in many urban areas with dense populations despite a reduction of $NO_x$ concentrations at all sites. Through detailed photochemical modelling, isopleth plots for $O_3$ formation rates are constructed. Visualization of year-to year variations per station type using two seasonal isopleth plots allows for assessing the effectiveness of precursor controls in Saxony over the past 20 years and offers insights into potential future $O_3$ pollution mitigation strategies.

It is shown that the $O_3$ formation dynamics across traffic and urban background sites were determined to be predominantly VOC-limited from 2000 to 2019. The observed increases in NetPO$_3$ and $O_3$ levels affirm that current efforts to reduce TNMVOC emissions from various sources are still insufficient. Continuing with a similar magnitude of VOCs reductions in densely populated regions over the next years will likely result in further deterioration of $O_3$ pollution rather than its mitigation. Based on anthropogenic and biogenic emission data, we suggest that moderate $NO_x$ reduction and additional VOCs emission controls should be implemented, with particular attention given to solvent emissions, in order to more effectively alleviate regional $O_3$ formation. Given the detailed effects of various NMVOCs emissions on $O_3$ increases and the complete lack of comprehensive NMVOCs measurements, it is also strongly recommended to implement such monitoring. This would be of great help to assess emission inventories and improve the robustness of modelling studies on $O_3$ formation, not at least to develop prediction capability and undertake scenario calculations as to which path atmospheric ozone pollution will follow.

**Data availability.** The measurement data used in this study is freely available from the LfULG (https://www.umwelt.sachsen.de/umwelt/infosysteme/luftonline/recherche.aspx) or can be made available upon request to the corresponding author.

**Supplement.** The supplement related to this article is available online at:



**Author contributions.** SB, DvP, AT and HH conceptualized the study. YW with support from DvP, AT and EHH curated the data, performed the formal analyses and visualized the results. MH provided the VOCs measurements. HH supervised the entire study. YW drafted the manuscript, which all authors have reviewed and edited.

**Competing interests.** The contact author has declared that none of the authors has any competing interests.

**Acknowledgements.** We appreciate the operation of the monitoring network, data provision and logistic support at the Borna station by the Saxonian "Betriebsgesellschaft für Umwelt und Landwirtschaft" (BfUL).

**Financial support.** This research has been funded through the SAXOZONE project (grant no. 51-Z266/20) by the Saxonian State Office for the Environment, Agriculture and Geology (LfULG). Certain aspects of the ozone analysis work have also been supported through the DFG project "Coupling and Abatement of atmospheric Ozone and PM in the Chinese Yangtze River Delta (PMO3)" under HE3086/46-1 with project No 448587068.

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
