# Peer review of "Ozone (O3) observations in Saxony, Germany for 1997 - 2020: Trends, modelling and implications for O3 control"

_EGUsphere, 2024_

## Author Response (AR1)

Dear Reviewer,

We sincerely appreciate the valuable and constructive feedback from Reviewer #1, which has significantly contributed to improving our manuscript. Below, we provide a detailed, point-by-point response to all comments. The reviewer's comments are presented in *italic black*, our responses in blue, and the corresponding revisions in the manuscript and Supporting Information are highlighted in red.

Thank you for your time and consideration.

All line numbers given in our below replies refer to the revised manuscript (color-tracked) and Supporting Information (color-tracked).

*General comments*

*The paper analyses the effect of NOx and VOC emission reductions on surface ozone using observations and a model in Saxonia. The paper fits to the scope of ACP. There are mostly technical issues that have to be fixed before publication like missing definitions or information on input data and model version, and the need to improve figures and tables.*

We thank the reviewer for his/her positive overall assessment of our manuscript.

*Specific comments*

**R1-C1:** *Line 12: Is NOx=NO+NO$_2$ or are also NO$_3$ and N$_2$O$_5$ included? Please define for clarity.*

**A1:** Here the term "oxides of nitrogen" (NO$_x$) only includes nitric oxide (NO) and nitrogen dioxide (NO$_2$). In the manuscript, a clarified definition of "oxides of nitrogen (NO$_x$ = NO+NO$_2$)" has been added in the line 12 and now reads:

The findings indicate that despite reductions in oxides of nitrogen (NO$_x$ = NO+NO$_2$) concentrations across all sites, O$_3$ pollution in Saxony has in fact worsened over the past decade, especially in densely populated urban areas.

According to ACP's submission requirements, abbreviations must be defined at their first occurrence in both the abstract and the main manuscript. So "= NO+NO$_2$" was added at line 30 of the manuscript as well, the phrase now reads:

oxides of nitrogen (NO$_x$ = NO+NO$_2$)

**R1-C2:** *Line 16: Spell out "VOC" already here, line 29 (or line 19, hidden) is too late.*

A2: The definition of VOCs now appears earlier in the manuscript. "Volatile organic compounds (VOCs)" was added to line 17 of the manuscript, the revised sentence now reads:

To diagnose O$_3$ formation and the controlling effects of NO$_x$ and volatile organic compounds (VOCs) over the past decades in this region, for the first time, detailed photochemical box modelling was performed by means of the complex MCM (Master Chemical Mechanism).

In line 31 of manuscript, the phrase of "volatile organic hydrocarbons" has been corrected to "volatile organic compounds".

The abstract must be under 250 words per submission guidelines. To meet this requirement, some words were removed or replaced in lines 10-11 of the article. The revised sentences are as follows.

Given its importance for human health, vegetation, and climate, trends in ground-level ozone ($O_3$) concentrations in eastern Germany were systematically analysed using the long-term $O_3$ data from 16 measurement stations.

**R1-C3:** *Line 125: I suppose "p" is probability. Please spell out since "p" can be used also for pressure.*

A3: p-values means probability. Now "(probability)" has been added in the line 127 of the manuscript and the sentence reads:

Theil-Sen function derives p-values (probability) and uncertainties by bootstrap simulations.

**R1-C4:** *Line 131: Is this a boxmodel on trajectories? Please more details, from the given reference that is possible. The information on meteorology provided in the following lines is too general.*

A4: SPACCIM is used in the present study as box model (box fixed at a certain location). However, SPACCIM can be also used as air parcel model with changing conditions along a predefined trajectory. In the present study, temperature, pressure, and relative humidity are fixed parameters based on observed conditions.

A sentence describing the SPACCIM model has been added to lines 133-135 of the manuscript and "box model" has been indicated in line 133, now the revised sentences are shown below.

To understand the role of photochemistry for $O_3$ concentration evolution in Saxony, photochemical simulations were performed with the air parcel box model SPACCIM (SPectral Aerosol Cloud Chemistry Interaction Model). SPACCIM combines a size-resolved multiphase chemistry model with a microphysical model, enabling both to function independently while accounting for their interdependencies.

Some specific information on meteorology has been included in lines 149-152 of the article, and some sentences have been revised from line 146 to line 152. They now read:

Besides emission values, other initial parameters had to be adjusted to their typical daytime and nighttime levels under rural conditions (see Table S4 for details). For meteorological parameters and trace gas concentrations (except $SO_2$, HONO and PAN), data was derived from measurements in Sect. 2.1. The air temperature was set to 15°C in summer and 4°C in winter. The pressure was kept constant at 1000 hPa for both seasons, while the relative humidity was maintained at 70%. Ratio of solar radiation, defined as the mean value between 10:00 and 13:00 divided by the maximum clear sky radiation value during the same period, was calculated to 0.7 in summer and 0.4 in winter.

**R1-C5:** *Line 133: Which version of CAPRAM? You need it on the website cited in the supplement.*

A5: In the present study, only the detailed near-explicit gas-phase chemistry mechanism MCM v3.3.1 (Master Chemical Mechanism) is applied (this information can be found in the lines 138-139 of the manuscript). No version of CAPRAM (Chemical Aqueous Phase Radical Mechanism) is used. Although the SPACCIM model framework allows for the coupling of MCM with CAPRAM to simulate multiphase chemistry, it also enables the use of each mechanism independently. Here, SPACCIM is employed solely with the gas-phase chemistry mechanism MCM v3.3.1.

To avoid misunderstanding, the revised paragraph describing the mechanisms used in lines 132-139 of the text is as follows: To understand the role of photochemistry for $O_3$ concentration evolution in Saxony, photochemical simulations were performed with the air parcel box model SPACCIM (SPectral Aerosol Cloud Chemistry Interaction Model). SPACCIM combines a size-resolved multiphase chemistry model with a microphysical model, enabling both to function independently while accounting for their interdependencies. Detailed descriptions of the SPACCIM model framework can be found in Wolke

et al. (2005). In the present study, only the detailed near-explicit gas-phase chemistry mechanism, MCMv3.3.1, is used, which comprises 17224 reactions (http://mcm.york.ac.uk/MCM/)(Saunders et al., 2003).

Additionally, one reference (Saunders et al., 2003) has been inserted in lines 879-881 of the manuscript. The reference now reads:

Saunders, S. M., Jenkin, M. E., Derwent, R. G., and Pilling, M. J.: Protocol for the development of the Master Chemical Mechanism, MCM v3 (Part A): tropospheric degradation of non-aromatic volatile organic compounds, Atmospheric Chemistry and Physics, 3, 161-180, 2003.

**R1-C6:** *Line 141: After "S1" "showing NOx emissions of the main roads and urban centers".*

A6: In the manuscript, the phrase of "showing $NO_x$ emissions of the main roads and urban centres" has been added in lines 145-146, it now reads:

(see Fig. S1 showing $NO_x$ emissions of the main roads and urban centers)

**R1-C7:** *Line 147: The fixed value for HONO is not far off from typical values but it might be better to keep HONO/NOx constant which might help to get a faster increase of NO due to HONO photolysis after sunrise in Fig. 8, especially in winter (Elshorbany et al. 2012). Maybe for a sensitivity study or outlook.*

A7: Our model is initialized by setting the pollutant concentrations. Usually, we do not constrain concentrations based on measured time series. This artificial temporal fixation of the model is not done in our group because we want also to understand current model limitations which gets mask by constraining certain key species. Moreover, with the current model framework, it is difficult to maintain a constant $HONO/NO_x$ ratio throughout the simulation. Nevertheless, we think the idea to keep one compound such as HONO constant might be a way to identify missing sources/sinks and could be a valuable task for future sensitivity studies.

The authors acknowledge that this is a very interesting point, and we will keep it in mind for further model development.

**R1-C8:** *Line 166: Does table S4 refer to the pre-runs?*

A8: No, Table S4 refers to the prepared initial concentrations for further 24-hour simulations, which are defined as the base case simulations. Besides, as a result of adding one new table (Table S3) showing the emission data (see R1-C14), the numbering of original Table S4 now is revised to Table S5.

The revised text in lines 174-176 of the article is now:

The output concentrations from the last day of the pre-run simulations were used as the initial and boundary chemical data for the final 24-hour modelling. Dominant initial concentrations are given in Table S5 in the Supporting Information.

**R1-C9:** *Line 255: Isn't Fichtelberg due to the prevailing wind direction less affected by air masses from polluted regions than the other mountain stations?*

A9: We thank the reviewer for this insightful comment to make us gain a better understanding of ozone trends at Fichtelberg.

The following figure presents wind rose plots at different mountain sites over the past 10 year (2011-2020), generated using the *windRose* function of the package *openair* (Carslaw and Ropkins, 2012) in R (R core Team, 2020). At Fichtelberg, wind direction is predominantly westerly, accompanied by relatively strong wind speeds. A similar prevailing wind direction is

observed at Carlsfeld, though with lower wind speeds. Both sites are indeed less affected by polluted air masses than the other two mountain sites Schwartenberg and Zinnwald, which have higher frequencies of air masses from the South and South-East (Czech Republic). However, in contrast to Carlsfeld, with a similar wind rose, only Fichtelberg observed the strongly decreasing trend over the past 10 years. We therefore believe -as already discussed in the original manuscript- that it might be the higher impact of free tropospheric air masses at Fichtelberg, which makes its trend differ from the lower-altitude sites in the region.

[Figure]

**R1-C10:** *Figure 4, caption: Better write "differences" as in the text.*

A10: The caption of Fig. 4 has been revised. See lines 275-277 in the article:

Figure 4: Mean yearly $O_3$ concentration differences, i.e. decrements, between different station types. Yellow point lines represent decrements from mean mountain to mean rural stations concentrations, blue point lines represent rural to urban, and red point lines represent urban to traffic stations decrements.

**R1-C11:** *Line 266: From Fig. 4 it can be also said that the difference between rural and urban stays about constant if the first 2 points are included. Please improve text in this paragraph.*

A11: We thank the reviewer for this useful comment. The reviewer is right. The difference between rural and urban indeed stays about constant if the first 2 points in 1997-1998 are considered.

Now the text in this paragraph (at lines 281-294) has been revised, the paragraph now reads:

The urban decrement, i.e. rural to urban background difference, remained rather stagnant around 7 µg m$^{-3}$ in all years except for the period 1999-2003. The traffic decrement from the urban background to the one long-term $O_3$ traffic station DD-Nord also decreased from about 15 to about 5 - 7 µg m$^{-3}$ and is now at a similar level as the urban decrement. If the $O_3$ trends mentioned above continue in a similar way in the future, it can be expected that the typical $O_3$ concentrations at traffic stations will become more similar to those in the urban background and rural background concentrations will become increasingly similar to those at the mountain sites. The Air Quality Expert Group (2021) reported a gradual convergence of urban and rural $O_3$ levels in the UK from 2000 to 2019, which is in contrast to the stable difference in $O_3$ concentration between urban and rural background observed since 2004 in the present study. The reason for this is the similarly increasing $O_3$ trends at urban and rural background sites in Saxony (Fig. 3), keeping the urban to rural decrement roughly similar. In any case, due to the typical settlement densities, these trends would mean higher $O_3$ exposures for the vast majority of the population.

**R1-C12:** *Lines 340f: These numbers do not agree with Fig. 5, 100, 50 and 0 percentiles. Typos or misunderstanding? Please correct or clarify.*

A12: The reviewer is right, and these were incorrect values. We much appreciate his/her attention to the details! The main issue was in the R code the authors used to calculate the averages. we mistakenly filtered the data using "==" instead of "%in%." This led to discrepancies in the averaged results across different site types and percentiles, even when some values appeared similar. We had confused the two operators in the code.

The authors have updated the data in the text by recalculating the averages using the correct code. The corrected data can be found in lines 334 through 373, as shown below:

[revised manuscript text omitted]

***R1-C13:*** *Line 362: Insert "of the latest period"*

A13: The phrase "of the latest period" has been added to lines 386-387 of the article. Now the sentence read:

For the traffic site, increasing trends are observed in all seasons over all time periods, but strongest in summer or spring of the latest period.

***R1-C14:*** *Line 431: A table with emissions for this base case for Saxony would be useful here or earlier (line 141). Or refer at least to Figs. 11 and S9.*

A14: A summary table, Table S3, presenting the annual emission data in 2019 for the entire Saxony region in summer and winter simulation scenarios, has been added at line 27 of the Supporting Information. Given the large space needed to include this additional information, we now give the Table as a separate Excel file and include only the Table header in the Supporting information pdf file. See below:

Table S3. The annual emission data in 2019 for the entire Saxony region in summer and winter simulation scenarios. Each species is given by its SMILES string, and its IUPAC name generated using the PubChemPy package in Python (https://pubchempy.readthedocs.io/en/latest/), along with the corresponding chemical structure. It is noted that for some compounds, IUPAC names could not be retrieved from the PubChem database by the tool and they are therefore left blank.

See attached Excel file named *Table S3_The annual emission data in 2019 for the entire Saxony region.*

As a result of adding one new table, the numbering of subsequent tables in the Supporting Information (starting from Table S3) has been updated accordingly.

Additionally, the phrase of "(Table S3)" has been inserted in line 143 of the manuscript. The sentence now reads:

The summer and winter emission data (Table S3) was based on anthropogenic and biogenic emission inventories in 2019 from the German Environment Agency (UBA) for Germany and Thürkow et al. (2024), and derived for the whole Saxony area (50.9° N latitude and 14.3° E longitude) (see Fig. S1 showing $NO_x$ emissions of the main roads and urban centres).

***R1-C15:*** *Line 478: "of the base cases for Saxony (Sect 3.3.1)", right? If yes please insert.*

A15: Yes, right. The phrase of "of the base cases for Saxony (Sect 3.3.1)" has been inserted in line 517 of the article, as shown below:

These simulations of the base cases for Saxony (Sect 3.3.1) aimed to elucidate the $O_3$ production rate with regards to the ambient concentrations of $NO_x$ and TNMVOC, which is depicted in the ozone isopleths of Fig. 10.

***R1-C16:*** *Line 498: "for the station types"? Insert if yes, if no please explain already earlier in this subsection.*

A16: Yes, the phrase of "for the station types" has been inserted in line 557 of the article. The sentence, incorporating some other modifications in response to comments R2-C9, has been revised at line 556-557 of the manuscript. It now read:

For further clarification, in Tables S9 and S10, these TNMVOC estimates are shown together with a comparison of measured and modelled $NO_x$ and $dO_3/dt$ for the station types.

**R1-C17:** *Line 503: Mention the mean values also here, then table S11 is not needed.*

A17: Mean values have now been added in lines 560-563 of the article, and Table S11 in the Supporting Information has been deleted. The revised sentence in the article is shown below:

Indeed, unpublished data for a range of NMVOCs observed throughout the year 2022 in Borna, a city south of Leipzig, exhibited remarkably low concentrations of 66 NMVOCs species (Table S11) at a near-road measurement site, with hourly mean and maximum total mixing ratios of 3.6 ppb and 29.7 ppb in summer and 6.0 ppb and 204.6 ppb in winter, respectively.

Note that in the revised sentences above, that Table S11 corresponds to the original Table S10. Due to the addition of a new table (Table S3) showing the emission data (see R1-C14), the numbering of subsequent tables in the Supporting Information (starting from Table S3) has been updated accordingly.

***Technical corrections***

**R1-C18:** *Line 148: Better new paragraph.*

A18: Due to the added meteorology descriptions in lines 149-152 of the manuscript, the authors believe it would be better to start the new paragraph from "The initial $SO_2$..." instead of "CO and $CH_4$…".

So, a new paragraph starting with "The initial $SO_2$..." has already been added at line 153 of the article.

**R1-C19:** *Fig. 1: Include the frame of Fig. S1 for convenience.*

A19: A newly updated map with a frame that approximates the spatial extent of the state of Saxony (Fig. S1) has been added to line 18 of the Supporting Information. Additionally, some modifications have been made based on R1-C25. See the updated Fig. S1 below:

[Figure]

*R1-C20: Fig. 2: Include in caption after "1997" "or later".*

A20: The phrase "or later" has been added after "1997" has been added in line 223 of the article. Now the revised caption read: Figure 2: Boxplots of $O_3$ concentrations at individual monitoring stations, coloured according to their station type. The middle yellow point and the black horizontal line indicate the mean and median, respectively. The ends of the box indicate the lower and upper quartiles, the antennas the 1.5-fold interquartile range (IQR) and individual points are extreme values outside the IQR. Data per station from 1997 or later onward to 2020. The detailed station data are summarised in Table S1 in the Supporting Information.

*R1-C21: Fig. 5: Here colors superfluous.*

A21: Although the colors are somewhat excessive, the authors prefer the original figure. There is also a version that uses a single color to represent dots at different percentiles, but dots appear a bit crowded, making it difficult to distinguish between percentiles. See below:

[Figure]

*R1-C22: Lines 665, 673, 700: Please provide DOI or URL for technical reports.*

A22: URL links have been added in lines 730-731, 740, and 767 of the article. Now the references read:

Colette, A., Beauchamp, M., Malherbe, L., and Solberg, S.: Air quality trends in AIRBASE in the context of the LRTAP Convention, ETC/ACM Technical Paper, 4, 2015, https://www.eionet.europa.eu/etcs/etc-atni/products/etc-atni-reports/ETCACM_TP_2015_4_AQtrends, 2015b.

Corsmeier, U., Kalthoff, N., Vogel, B., Hammer, M.-U., Fiedler, F., Kottmeier, C., Volz-Thomas, A., Konrad, S., Glaser, K., and Neininger, B.: Ozone and PAN formation inside and outside of the Berlin plume—Process analysis and numerical process simulation, Tropospheric Chemistry: Results of the German Tropospheric Chemistry Programme, 289-321, https://doi.org/10.1007/978-94-010-0399-5_13, 2002.

EEA: European Union emission inventory report 1990-2021 — Under the UNECE Convention on Long-range Transboundary Air Pollution (Air Convention), Report 04/2023, https://data.europa.eu/doi/10.2800/68478, 2023.

*R1-C23: Lines 712, 717, 747, 789, 794: Same journal? If yes use the same abbreviation.*

A23: Yes. The journal name "Elementa: Science of the Anthropocene" has been corrected in lines 779 and 784 of the article. Now the references read:

Fleming, Z. L., Doherty, R. M., Von Schneidemesser, E., Malley, C. S., Cooper, O. R., Pinto, J. P., Colette, A., Xu, X., Simpson, D., and Schultz, M. G.: Tropospheric Ozone Assessment Report: Present-day ozone distribution and trends relevant to human health, Elementa: Science of the Anthropocene, 6, 12, 2018.

Gaudel, A., Cooper, O. R., Ancellet, G., Barret, B., Boynard, A., Burrows, J. P., Clerbaux, C., Coheur, P.-F., Cuesta, J., and Cuevas, E.: Tropospheric Ozone Assessment Report: Present-day distribution and trends of tropospheric ozone relevant to climate and global atmospheric chemistry model evaluation, Elementa: Science of the Anthropocene, 6, 2018.

*R1-C24: Line 772: Final revised version available?*

A24: Yes, the final revised version is available. The reference for the final revised version has been updated in lines 849-852 of the manuscript. The reference now reads:

Nussbaumer, C. M., Crowley, J. N., Schuladen, J., Williams, J., Hafermann, S., Reiffs, A., Axinte, R., Harder, H., Ernest, C., Novelli, A., Sala, K., Martinez, M., Mallik, C., Tomsche, L., Plass-Dülmer, C., Bohn, B., Lelieveld, J., and Fischer, H.: Measurement report: Photochemical production and loss rates of formaldehyde and ozone across Europe, Atmospheric Chemistry and Physics, 21, 18413-18432, 2021.

***R1-C25:*** *Supplement: Fig. S1: Please define grid and its relation to the one in Fig. 1. It appears to be a subset of Fig. 1. Meaning or purpose of red rectangles not clear (is the sum of all 3 used in Fig. S9 which should be referred to?)? Mark the location of the extreme maximum which is almost one order of magnitude out of scale. Mark also the reference point mentioned in main text. Maybe a logarithmic color bar for the emissions would be useful.*

A25: In the original Fig. S1, three red rectangles in Fig. S1 approximated the entire Saxony area. However, it was three rectangular areas merely for technical reasons, as the averaged emission data could only be calculated for rectangles with the tool at hand.

To avoid confusion, an updated Fig. S1 is provided at line 18 of the Supporting Information, in which the approximate boundary of Saxony is marked only by using a continuous red frame instead of three separate rectangles. This red frame indicates the simulated entire Saxony area. See the updated Fig. S1 below.

[Figure]

To clarify the relationship between Fig. 1 and Fig. S1, the boundary of Saxony has also been outlined in the updated Fig. 1. The updated Fig. 1 has been added at line 110 of the manuscript, as shown below.

[Figure]

By comparing the updated Fig. 1 and Fig. S1, we can see the outlined Saxony region (red line) in Fig. S1 roughly corresponds to the shape of the Saxony boundary shown in Fig. 1.

The emissions data in each year, illustrated in Fig. 11 in the manuscript and Fig. S9 in the Supporting Information, are obtained by averaging the emissions values based on the red-outlined entire Saxony area in the updated Fig. S1.

Besides, the caption of Fig. 11 has been revised at line 644-646 of the manuscript, as shown below:

Figure 11: Anthropogenic emissions of NMVOCs in Saxony for five years from 2000-2019. Emission data were obtained by averaging values across the approximated area of the entire state of Saxony (see Fig. S1). Details of the emission categories and corresponding emission sectors can be found in Table S13.

The caption of Fig. S9 has been revised at line 138-140 of the Supporting Information, Due to the addition of two new figures (as Fig. S6 and S7, referring to R2-C9), the numbering of subsequent figures in the Supporting Information (starting from Fig. S6) has been updated accordingly. So the numbering of Fig. S9 should be Fig. S11. See the revised caption of Fig. S11 below.

Figure S11: Anthropogenic emissions of $NO_x$ in Saxony for five years from 2000-2019. Emission data were obtained by averaging values across the approximated area of the entire state of Saxony (see Fig. S1). Details of the emission categories and corresponding emission sectors can be found in Table S13.

About the reviewer's comment on the maximum and minimum values in the original Fig. S1, now the authors make the following response to explain.

The maximum and minimum values initially reflected extreme high and low values in whole Germany in the original Fig. S1. The extreme low and high values were located in other regions of Germany rather than Saxony. In the updated Fig. S1, the authors have adjusted the color pie chart to show only the data range for Saxony in a non-logarithmic scale. It is clear from the updated Fig. S1 where the location of the maximum value (~0.015 kg m$^{-2}$) for the state of saxony is located.

In the updated Fig. S1, except for the red-outlined entire Saxony area, two specific rectangles in rose pink, A and B, represent the representative urban area (city of Leipzig) and traffic-dominated area (highway close to Dresden) used for the simulation, respectively. Now the caption of Fig. S1 has been revised in the Supporting Information. See below:

Figure S1: The $NO_x$ emission (in kg m$^{-2}$) from road transport in the entire Saxony state region in 2019. A continuous red frame indicates the entire simulated Saxony area. Two specific rectangles in rose pink, A and B, represent the representative urban

area (city of Leipzig) (0.13° × 0.20°) and traffic-dominated area (highway close to Dresden) (0.07° × 0.10°) used for the simulation, respectively.

One sentence has been revised in lines 143-146 of the manuscript, now it reads:

The summer and winter emission data (Table S3) was based on anthropogenic and biogenic emission inventories in 2019 from the German Environment Agency (UBA) for Germany and Thürkow et al. (2024), and derived for the whole Saxony area (50.9° N latitude and 14.3° E longitude) (see Fig. S1 showing $NO_x$ emissions of the main roads and urban centres).

**R1-C26:** *Table S3: "Dry deposition", right? It might be useful to include conversion factors to ppbv for the species not listed with this unit.*

A26: Yes, the reviewer is right. "Deposition rate" has now been revised to "Dry deposition velocity".

The concentration unit in the updated table is uniformly set to ppb.

Additionally, a typo has been corrected, "$HSO_4$" is now "$H_2SO_4$". The concentrations of $SO_2$ and $CH_4$ were incorrect due to an error in the conversion from molec $cm^{-3}$ to µg $m^{-3}$, caused by an incorrect conversion factor, which has now been corrected. The concentrations used in the modelling, expressed in molec $cm^{-3}$, are provided in Table S5 (former Table S4) for reference. The concentration of $SO_2$ should be 3 µg $m^{-3}$ (1.1 ppb) in summer and 5 µg $m^{-3}$ (1.9 ppb) in winter, instead of the previously written 54 µg $m^{-3}$ in summer and 3 µg $m^{-3}$ in summer. The concentration of $CH_4$ in summer and winter should be 2018.6 µg $m^{-3}$ (1700 ppb) instead of 1155.8 µg $m^{-3}$. The originally cited reference Schaefer (2019) has been replaced with Herrmann et al. (2000).

The definition of ratio of solar radiation has been revised for clarity and accuracy. As a result of adding one new table (Table S3) showing the emission data (see R1-C14), the numbering of original Table S3 now is revised to Table S4. Now Table S4 shows as:

| Model setting | Unit | Summer | Winter | Reference |
|---|---|---|---|---|
| Initial time | | 00:00 CET, 14 July, 2019 | 00:00CET, 14 January, 2019 | |
| Meteorological conditions | | | | |
| Temperature | °C | 15 | 4 | Measured |
| Pressure | hPa | 1000 | 1000 | Measured |
| Relative humidity | % | 70 | 70 | Measured |
| Ratio of Solar radiation* | | 0.7 | 0.4 | Measured |
| Dry deposition velocity | | | | |
| $NO_2$ | cm $s^{-1}$ | 0.3 | 3 | Rondón et al. (1993) |
| $N_2O_5$ | cm $s^{-1}$ | 100 | 2 | Hoffmann et al. (2019) |
| $O_3$ | cm $s^{-1}$ | 0.8 | 0.08 | Clifton et al. (2020) |
| NO | cm $s^{-1}$ | 0.05 | 0.05 | Zhu et al. (2020) |
| $HNO_3$ | cm $s^{-1}$ | 3.5 | 3.5 | Zhu et al. (2020) |

| | | | | |
|---|---|---|---|---|
| $H_2O_2$ | cm s$^{-1}$ | 1 | 1 | Zhu et al. (2020) |
| CO | cm s$^{-1}$ | 0.1 | 0.1 | Zhu et al. (2020) |
| HCl | cm s$^{-1}$ | 1 | 1 | Zhu et al. (2020) |
| $NH_3$ | cm s$^{-1}$ | 1 | 1 | Zhu et al. (2020) |
| $SO_2$ | cm s$^{-1}$ | 1 | 1 | Zhu et al. (2020) |
| $H_2SO_4$ | cm s$^{-1}$ | 2 | 2 | Zhu et al. (2020) |
| HCHO | cm s$^{-1}$ | 1 | 1 | Zhu et al. (2020) |
| $CH_3OH$ | cm s$^{-1}$ | 1 | 1 | Zhu et al. (2020) |
| $CH_3CH_2OH$ | cm s$^{-1}$ | 0.5 | 0.5 | Zhu et al. (2020) |
| PANs | cm s$^{-1}$ | 0.7 | 0.7 | Wu et al. (2012) |
| CHClO | cm s$^{-1}$ | 0.2 | 0.2 | Hoffmann et al. (2019) |
| CHBrO | cm s$^{-1}$ | 0.2 | 0.2 | Hoffmann et al. (2019) |
| **Boundary layer heights (BLHs)** | | | | |
| Daytime BLHs | m | 500 | 2000 | 2 |
| Nighttime BLHs | m | 250 | 1000 | |
| **Measured Chemical data** | | | | |
| $O_3$ | ppb | 31.7 | 21.6 | Measured |
| $NO_2$ | ppb | 3.4 | 7.0 | Measured |
| $SO_2$ | ppb | 1.1 | 1.9 | UBA website |
| CO | ppb | 153.4 | 153.4 | Zellweger et al. (2009) |
| $CH_4$ | ppb | 1700 | 1700 | Herrmann et al. (2000) |
| HONO | ppb | 0.5 | 0.5 | Stieger et al. (2018) |
| PAN | ppb | 0.5 | 0.5 | Pandey Deolal et al. (2014) |

\* Ratio of solar radiation, defined as the mean value between 10:00 and 13:00 divided by the maximum clear sky radiation value during the same period.

In the lines 153-155 of the manuscript, the sentences have been revised. Now they read:

The initial $SO_2$ concentrations in summer and winter were obtained from UBA (https://www.umweltbundesamt.de/daten). CO and $CH_4$ were set with 153.4 and 1700 ppb, respectively, referred the measured data from Zellweger et al. (2009) and Herrmann et al. (2000).

**R1-C27:** *Table S4: Your compound strings are often inconsistent with the MCM-notation and could not be found in the CAPRAM-link. Instead of listing them twice it would be useful to replace one column by a column with the full chemical notation or the compound names as in Table S10. Also some compound strings change from summer to winter, typos?*

A27: Yes, some SMILES strings are available on the MCM website, while others are not. Additionally, there is no publicly accessible webpage in the CAPRAM link for referencing these SMILES strings - they are only accessible to CAPRAM users.

Instead of listing the SMILSE strings twice, one column with the full SMILSE strings, one column with their IUPAC names and one column with their chemical structures are presented in the updated table.

The following table shows the difference between summer and winter species in the last 3 or 4 rows of the original Table S4. There are typos. The first three SMILES strings (shaded in gray) under the summer category should be removed in the updated table, as they are followed by four species listed for the winter.

| Compound string Summer | Unit molec cm$^{-3}$ | Compound string Winter | Unit molec cm$^{-3}$ |
|---|---|---|---|
| CC(=O)C(OO)C(O)(C)C | 2.31E+00 | CC(=O)CC(O)(C)CO | 7.43E+03 |
| CC(=O)CC(O)(C)COO | 1.08E+00 | CC(=O)CC(O)(C)C=O | 8.51E+02 |
| CC(=O)CC(O)(C)CON(=O)=O | 4.82E+00 | CC(O)(C)CC(=O)COO | 7.98E+05 |
| CC(=O)CC(O)(C)CO | 9.24E+03 | CC(O)(C)CC(=O)CO | 3.28E+04 |

As a result of adding one new table (Table S3) showing the emission data (see R1-C14), the numbering of original Table S4 now is revised to Table S5. The revised caption of Table S5 has been updated in lines 38-43 of the Supporting Information and now reads:

Table S5. Dominant initial gas-phase concentrations applied in the final 24-hour simulations for summer and winter scenarios. Each species is given by its SMILES string, and its IUPAC name generated using the PubChemPy package in Python (https://pubchempy.readthedocs.io/en/latest/), along with the corresponding chemical structure. It is noted that for some compounds, IUPAC names could not be retrieved from the PubChem database by the tool and they are therefore left blank.

See attached Excel file named *Table S5_Dominant initial gas-phase concentrations.*

**R1-C28:** *Table S5: Add in caption or text, line 176, "i.e. each combination is considered."*

A28: The original Table S5 may have caused some misunderstanding due to a lack of clarity regarding the changing emission factor for summer and winter simulations. To improve clarity, the authors have revised both the table and its caption.

The phrase "considering each combination" has been added to the caption of Table S6 (former S5) at lines 46-48 in the Supporting Information. Table S6 now reads:

Table S6. Changing the emission multiplier for $NO_x$ and TNMVOC in the simulations. Three batches were conducted, considering each combination, with each batch comprising 400 model runs. The total number of simulations is 800 for summer and 1200 for winter.

| Batch 1 (Summer/Winter) | | | Batch 2 (Summer/Winter) | | | Batch 3 (Winter) | | |
|---|---|---|---|---|---|---|---|---|
| Number | Emission factor | | Number | Emission factor | | Number | Emission factor | |
| | $NO_x$ | TNMVOC | | $NO_x$ | TNMVOC | | $NO_x$ | TNMVOC |

| | | | | | | | | | |
|---|---|---|---|---|---|---|---|---|---|
| 1 | 0.001 | 0.001 | 1 | 1.5 | 1.5 | 1 | 1.5 | 13 |
| 2 | 0.005 | 0.005 | 2 | 2 | 2 | 2 | 2 | 14 |
| 3 | 0.01 | 0.01 | 3 | 2.5 | 2.5 | 3 | 2.5 | 16 |
| 4 | 0.05 | 0.05 | 4 | 3 | 3 | 4 | 3 | 17 |
| 5 | 0.1 | 0.1 | 5 | 3.5 | 3.5 | 5 | 3.5 | 18 |
| 6 | 0.5 | 0.5 | 6 | 4 | 4 | 6 | 4 | 19 |
| 7 | 1 | 1 | 7 | 4.5 | 4.5 | 7 | 4.5 | 21 |
| 8 | 5 | 5 | 8 | 5.5 | 5.5 | 8 | 5.5 | 22 |
| 9 | 10 | 10 | 9 | 6 | 6 | 9 | 6 | 23 |
| 10 | 15 | 15 | 10 | 6.5 | 6.5 | 10 | 6.5 | 24 |
| 11 | 20 | 20 | 11 | 7 | 7 | 11 | 7 | 26 |
| 12 | 25 | 25 | 12 | 7.5 | 7.5 | 12 | 7.5 | 27 |
| 13 | 30 | 30 | 13 | 8 | 8 | 13 | 8 | 28 |
| 14 | 35 | 35 | 14 | 8.5 | 8.5 | 14 | 8.5 | 29 |
| 15 | 40 | 40 | 15 | 9 | 9 | 15 | 9 | 30 |
| 16 | 45 | 45 | 16 | 9.5 | 9.5 | 16 | 9.5 | 31 |
| 17 | 50 | 50 | 17 | 10.5 | 10.5 | 17 | 10.5 | 32 |
| 18 | 60 | 60 | 18 | 11 | 11 | 18 | 11 | 33 |
| 19 | 70 | 70 | 19 | 11.5 | 11.5 | 19 | 11.5 | 34 |
| 20 | 80 | 80 | 20 | 12 | 12 | 20 | 12 | 35 |
| **Total simulation number** | **400** | | **400** | | | **400** | | |

One sentence has been revised in lines of 186-189 in the manuscript, now it reads:

The sensitivity simulations were done by scaling the base case emissions of TNMVOC and $NO_x$ 20 times in each of three batch runs, i.e. each combination is considered (Table S6). Three batches were performed to achieve a sensible range of resulting TNMVOC and $NO_x$ concentrations in the total of 800 and 1200 model runs for summer and winter, respectively.

***R1-C29:*** *Fig. S2: There is enough space to keep the time axis in the last frame on scale.*

A29: The time axis in the last frame has been adjusted in the updated Fig. S2. See below:

[Figure]

*R1-C30: Fig. S9 to S13 and 11: It is often difficult to attribute the legends with source categories to the color bars in the figures since the order differs from figure to figure. Please stay to the order in the legends in every figure. Captions too short or misleading (e.g. traffic dominated by solvents in Fig. S10).*

A30: To enhance readability, the legends (in Fig. 11 in the manuscript and Figs. S9-S13 in the Supporting Information) have been arranged in descending order of emissions percentage.

Since each emission category includes lots of emission sectors, it is impractical to write all the emission sectors clearly in the caption of the figures. Here a new table (as Table S13) has been added at line 183 of the Supporting Information to show the emission categories and emission sectors in detail. Now the caption of Table S13 reads:

Table S13: Emission categories and corresponding emission sectors (Schneider et al., 2016).

See attached Excel file named *Table S13_Emission categories and corresponding emission sectors.*

Here is an explanation regarding the reviewer's comment on the misleading aspects of Fig. S10. Although the selected traffic area (indicated by the rectangular B in Fig. S1) is close to the roadside, if you zoom in the B area, you will see that there are also residents along roadside, and the solvents emitted are likely associated with domestic solvent use.

Anyway, the revised figure captions and figures are shown below. It is noted that, due to the addition of two new figures (as Fig. S6 and S7, referring to R2-C9), the numbering of subsequent figures in the Supporting Information (starting from Fig. S6) has been updated accordingly. As a result, the numbering of Fig. S9 has been changed to Fig. S11, and the numbering of all following figures has been adjusted accordingly.

[Figure]

Figure 11: Anthropogenic emissions of NMVOCs in Saxony for five years from 2000-2019. Emission data were obtained by averaging values across the approximated area of the entire state of Saxony (see Fig. S1). Details of the emission categories and corresponding emission sectors can be found in Table S13.

[Figure]

Figure S11: Anthropogenic emissions of $NO_x$ in Saxony for five years from 2000-2019. Emission data were obtained by averaging values across the approximated area of the entire state of Saxony (see Fig. S1). Details of the emission categories and corresponding emission sectors can be found in Table S13.

[Figure]

Figure S12: Anthropogenic emissions of NMVOCs in traffic area for five years from 2000-2019. Emission data were obtained by averaging values from the selected traffic-dominated area (see Fig. S1). Details of the emission categories and corresponding emission sectors can be found in Table S13

[Figure]

Figure S13: Anthropogenic emission of NO$_x$ in traffic area for five years from 2000-2019. Emission data were obtained by averaging values from the selected traffic-dominated area (see Fig. S1). Details of the emission categories and corresponding emission sectors can be found in Table S13.

[Figure]

Figure S14: Anthropogenic emission of NMVOCs in urban area for five years from 2000-2019. Emission data were obtained by averaging values from the selected urban area (city of Leipzig) (see Fig. S1). Details of the emission categories and corresponding emission sectors can be found in Table S13.

[Figure]

Figure S15: Anthropogenic emission NO$_x$ in urban area for five years from 2000-2019. Emission data were obtained by

averaging values from the selected urban area (city of Leipzig) (see Fig. S1). Details of the emission categories and corresponding emission sectors can be found in Table S13.

*R1-C31: Fig. S13 and S14: Better convert to a table.*

A31: The authors assume the reviewer are referring to Figs. S14 and S15. A new table (Table S12) summarizing the data from Figs. S14 and S15 has been added at line 181 of the Supporting Information. See below:

Table S12. A summary of the emission data from Figs. S16 (former Fig. S14) and S17 (former Fig. S15).

| Emission species | Emission source | Saxony | Leipzig-urban |
|---|---|---|---|
| | | Emission / kg m$^{-2}$ year $^{-1}$ | |
| Alpha-pinene | Biogenic emissions | 0.00098 | 0.00008 |
| Isoprene | Biogenic emissions | 0.00041 | 0.00005 |
| Limonene | Anthropogenic and biogenic emissions | 0.00073 | 0.00528 |

Additionally, the captions of Figs. S16 (former Fig. S14) and S17 (former Fig. S15) have been revised accordingly at lines 168-172 and 175-179 of the Supporting Information. They now read:

Figure S16: Biogenic emissions of isoprene (ISO) and alpha-pinene (API) in Saxony for the year 2019, along with anthropogenic and biogenic emission data for limonene (LIM) in the same year. Emission data were obtained by averaging values across the approximated area of the entire state of Saxony (see Fig. S1). The emission data source is from the German Environment Agency (UBA) for Germany and Thürkow et al. (2024). Detailed data can be referred to Table S12.

Figure S17: Biogenic emissions of isoprene (ISO) and alpha-pinene (API) in urban area for the year 2019, along with anthropogenic and biogenic emission data for limonene (LIM) in the same year. Emission data were obtained by averaging values from the selected urban area (city of Leipzig) (see Fig. S1). The emission data source is from the German Environment Agency (UBA) for Germany and Thürkow et al. (2024). Detailed data can be referred to Table S12.

And the phrase "and Table S12" has been added to lines 623-626 of the manuscript. The revised sentences now read:

Although the biogenic emissions data of year 2019 in overall Saxony (Fig. S16 and Table S12) are comparable to anthropogenic NMVOCs in same year (Fig. 11A), biogenic emissions of isoprene and alpha-pinene in selected urban stations (Fig. S17 and Table S12) are indeed several orders of magnitude smaller than the anthropogenic emissions data in these areas (Fig. S14A).

*R1-C32: Please no page break directly after the table head, and no line break in a number.*

A32: OK, in the manuscript (black version) and Supporting Information (black version), there is no page break directly after the table header, and numbers are kept intact without line breaks. Please check.


Dear Reviewer,

We sincerely appreciate the valuable and constructive feedback provided by Reviewer #2. These insights have significantly contributed to improving our manuscript.

Below, we provide a detailed point-by-point response to all comments. The reviewer's comments are highlighted in *italic black*, our responses are marked in blue, and the corresponding revisions in the manuscript and Supporting Information are indicated in red.

Thank you for your time and consideration.

All line numbers given in our replies given below refer to the revised manuscript (color-tracked) and Supporting Information (color-tracked).

*This paper summarizes the trends of O3 and its precursors over a 20 year period in Saxony and concludes that NOx reductions outpacing VOC reductions has led to increasing mean concentrations of ozone in urban regions. The paper fits the scope of ACP and is an important analysis with direct implications for designing effective future mitigation strategies in the region. Overall, the analysis is detailed despite the fact that there were a lot of assumptions that had to be made given data availability over this time period.*

We thank the reviewer for his/her positive overall assessment of our manuscript.

***General Comments:***

***R2-C1:*** *Overall, I found it difficult to read the paper given mis-matches between the text and figures/captions and I think particular attention needs to be paid to better align this information. In particular, the figure captions do not adequately describe was being shown in each figure and I was not able to determine what information was different in the subpanels without referencing the text. All figure captions should "stand alone" and be able to fully describe what is being shown and should be edited accordingly. Additionally, I found that there just seemed to be "missing" information where a figure was first referenced that was only mentioned much further down in the text. This made it difficult to understand what was in the figure the first time I encountered it, especially if that information was also not defined in the caption.*

(A) **A1:** We have carefully checked all Figure captions and revised the ones where we agree with the reviewer that more information needed to be given to make them stand alone. The substantially revised captions of Figures now read as follows: Caption changes in the main manuscript.

Figure 3: Boxplots of observed $O_3$ trends over three time periods until 2020 (given in columns 1-3) at the four station types. Transparent dots indicate statistically non-significant values. Black squares indicate means across all station values.

Figure 4: Mean yearly $O_3$ concentration differences, i.e. decrements, between different station types. Yellow point lines represent decrements from mean mountain to mean rural stations concentrations, blue point lines represent rural to urban, and red point lines represent urban to traffic stations decrements.

Figure 5: Theil-Sen trend values for different percentiles of $O_3$ concentrations in four station types (given in rows 1-4) and for three time periods until 2020 (given in columns A-C), respectively. Coloured dots show the values of individual stations, with solid dots for statistically significant trends and transparent dots otherwise. All dots are summarized by the boxplots. The black square reflects the mean trend value per percentile.

Figure 6: Theil-Sen trend values at different times of the year at the various station types (given in rows 1-4) and within each of the three time spans up to 2020 (given in columns A-C). Coloured dots show the values of individual stations, with solid dots for statistically significant trends and transparent dots otherwise. All dots are summarised by the boxplots. The black square shows the mean trend value per season.

Figure 7: Relationships between $O_3$ trends and $NO_x$ trends across all stations for the 3 different time periods (given in rows 1-3). Coloured dots show the values of individual stations, with solid dots for statistically significant $O_3$ trends and transparent dots otherwise. Also shown are linear fits for $O_3$ trends and $NO_x$ trends in different time periods.

Two newly updated Figs. 8 and 9, incorporating modifications in response to comments R2-C6 and R2-C7 have been added at lines 478 and 504 of the manuscript. See the revised captions of updated Figs. 8 and 9 below.

Figure 8: Diurnal profiles of hourly averaged modelled and observed NO and $NO_2$ in Saxony at rural background sites (A, C for summer case, and B, D for winter cases). Shaded areas indicate night-time, black lines indicate the modelled values, gray lines represent averaged observed concentrations and coloured lines refer to the observed concentrations at each station. It is noted that all observed NO concentrations below the DL of 1 $\mu g \ m^{-3}$ are not known and cannot be shown.

Figure 9: Diurnal profiles of hourly averaged modelled and observed $O_3$ in Saxony at rural background sites (A and B for summer and winter cases, respectively). Shaded areas indicate night-time, black lines indicate the modelled $O_3$, gray lines represent averaged observed $O_3$ and other coloured lines refer to the observed $O_3$ at each station.

A newly update Fig. 10 with some modifications referring to R2-C8, has been added at line 524 of the manuscript. In addition, to avoid the "missing" information caused by referencing a figure before it is discussed much further down in the text, the relevant details have also been included in the caption of Fig. 10. See the revised caption of updated Fig. 10 below.

Figure 10: Isopleth plots for the net $O_3$ production rate ($NetPO_3$ in ppb $h^{-1}$) during 12:00 - 13:00 CET as a function of averaged $NO_x$ and TNMVOC concentrations (A and B for summer and winter cases, respectively). $NetPO_3$ is shown using a rainbow color scale. The coloured points represent the conditions in different years with squares showing traffic stations and triangles showing mean urban stations conditions. The red dotted arrows represent hypothetical future scenarios with reductions in only one precursor (either TNMVOC or $NO_x$), while the yellow dotted arrows illustrate more plausible future pathways, where $NetPO_3$ slightly decreases through strong VOCs emission controls combined with moderate $NO_x$ reductions. Note that based on the underlying modelling, these data in Fig. 10 are given as mixing ratios. For better comparability with mass-based concentrations elsewhere in the manuscript, conversion factors of $NO_x$ ($\mu g \ m^{-3}$) $\approx 1.5 \times NO_x$ (ppb) and $O_3$ ($\mu g \ m^{-3}$) $\approx 2 \times O_3$ (ppb) can be used.

The caption of Figure 11 has been revised based on the comment of R1-C25 and R1-C30. See the revised version below.

Figure 11: Anthropogenic emissions of NMVOCs in Saxony for five years from 2000-2019. Emission data were obtained by averaging values across the approximated area of the entire state of Saxony (see Fig. S1). Details of the emission categories and corresponding emission sectors can be found in Table S13.

(B) Caption changes in the Supporting Information (SI)

A newly updated Fig. S1, incorporating modifications in response to comments R1-C19 and R1-C25, have been added at line 18 of the Supporting Information. See the revised captions of updated Fig. S1 and Fig. S3-S5 below.

Figure S1: The $NO_x$ emission (in kg $m^{-2}$) from road transport in the entire Saxony state region in 2019. A continuous red frame indicates the entire simulated Saxony area. Two specific rectangles in rose pink, A and B, represent the representative urban area (city of Leipzig) ($0.13° \times 0.20°$) and traffic-dominated area (highway close to Dresden) ($0.07° \times 0.10°$) used for the simulation, respectively.

Figure S3: Relationships between $O_3$ trends and NO, $NO_2$ trends across all stations for three different time periods (given in rows 1-3). Coloured dots show the values of individual stations, with solid dots for statistically significant $O_3$ trends and transparent dots otherwise. Also shown are linear fits for $O_3$ trends and NO, $NO_2$ trends in different time periods.

Figure S4: Linear correlations between measured averaged noon (12:00 - 13:00) $NO_x$ and $O_3$ in ppb of four station types in summer over 5 years (2000, 2005, 2010, 2015 and 2019). For better comparability with mass-based concentrations elsewhere in the manuscript, conversion factors of $NO_x$ (µg m$^{-3}$) $\approx 1.5 \times NO_x$ (ppb) and $O_3$ (µg m$^{-3}$) $\approx 2 \times O_3$ (ppb) can be used.

Figure S5: Linear correlations between measured averaged noon (12:00 - 13:00) $NO_x$ and $O_3$ in ppb of four station types in winter over 5 years (2000, 2005, 2010, 2015 and 2019). For better comparability with mass-based concentrations elsewhere in the manuscript, conversion factors of $NO_x$ (µg m$^{-3}$) $\approx 1.5 \times NO_x$ (ppb) and $O_3$ (µg m$^{-3}$) $\approx 2 \times O_3$ (ppb) can be used.

Two newly added figures as Fig. S6 and S7, in response to comments R2-C9, have been added at lines 98-107 of the Supporting Information. The numbering of subsequent figures in the Supporting Information (starting from Fig. S6) has been updated accordingly. The former Fig. S6 is revised to Fig. S8. See the revised captions of newly added Figs. S6 and S7 and Figs. S8-S10 below.

Figure S6: The distribution of resulting TNMVOC and $NO_x$ concentrations (in ppb) based on a total of 800 and 1200 model runs (see Table S6 for details) for summer (A) and winter (B), respectively. Panels C (summer) and D (winter) are subsets of Panels A and B, respectively, and use the same concentration coordinate ranges as Fig. S7. The different colors indicate different batch runs.

Figure S7: The grid of modelled net $O_3$ production rate (NetPO$_3$ in ppb h$^{-1}$) during 12:00 - 13:00 CET as a function of both modelled and emission inventory-derived $NO_x$ and TNMVOC concentrations (in ppb) (see Sect. 2.3) after bivariate linear interpolation. The X-axis and Y-axis in both summer (A) and winter (B) are interpolated using a very fine-resolved 1000 $\times$ 1000 grid. NetPO$_3$ is shown using a rainbow color scale.

Figure S8: Diurnal profiles of the modelled net $O_3$ production rate (NetPO$_3$ in ppb h$^{-1}$) and measured $O_3$ change rate (d$O_3$/dt, ppb h$^{-1}$ in Saxony at rural background sites (A and B for summer and winter cases, respectively). Shaded areas indicate night-time, black line indicates the modelled NetPO$_3$ and other coloured lines refer to the observed d$O_3$/dt at each station. Linear correlation coefficient (r) between modelled and measured rates is larger than 0.8 in both seasons.

Figure S9: Diurnal profiles of hourly averaged measured $O_3$ in summer over five years (2000, 2005, 2010, 2015, and 2019) across four station types.

Figure S10: Diurnal profiles of hourly averaged measured $O_3$ in winter over five years (2000, 2005, 2010, 2015, and 2019) across four station types.

Four newly updated Figs. S11 - S15, in response to comments R1-C30, have been added at lines 137-165 of the Supporting Information. See the revised captions of updated Figs. S11 - S15 below.

Figure S11: Anthropogenic emissions of $NO_x$ in Saxony for five years from 2000-2019. Emission data were obtained by averaging values across the approximated area of the entire state of Saxony (see Fig. S1). Details of the emission categories and corresponding emission sectors can be found in Table S13.

Figure S12: Anthropogenic emissions of NMVOCs in traffic area for five years from 2000-2019. Emission data were obtained by averaging values from the selected traffic-dominated area (see Fig. S1). Details of the emission categories and corresponding emission sectors can be found in Table S13.

Figure S13: Anthropogenic emission of NO$_x$ in traffic area for five years from 2000-2019. Emission data were obtained by averaging values from the selected traffic-dominated area (see Fig. S1). Details of the emission categories and corresponding emission sectors can be found in Table S13.

Figure S14: Anthropogenic emission of NMVOCs in urban area for five years from 2000-2019. Emission data were obtained by averaging values from the selected urban area (city of Leipzig) (see Fig. S1). Details of the emission categories and corresponding emission sectors can be found in Table S13.

Figure S15: Anthropogenic emission NO$_x$ in urban area for five years from 2000-2019. Emission data were obtained by averaging values from the selected urban area (city of Leipzig) (see Fig. S1). Details of the emission categories and corresponding emission sectors can be found in Table S13.

The captions of Figs. S16 and S17 has been revised based on the comment of R1-C31. See the revised versions below.

Figure S16: Biogenic emissions of isoprene (ISO) and alpha-pinene (API) in Saxony for the year 2019, along with anthropogenic and biogenic emission data for limonene (LIM) in the same year. Emission data were obtained by averaging values across the approximated area of the entire state of Saxony (see Fig. S1). The emission data source is from the German Environment Agency (UBA) for Germany and Thürkow et al. (2024). Detailed data can be referred to Table S12.

Figure S17: Biogenic emissions of isoprene (ISO) and alpha-pinene (API) in urban area for the year 2019, along with anthropogenic and biogenic emission data for limonene (LIM) in the same year. Emission data were obtained by averaging values from the selected urban area (city of Leipzig) (see Fig. S1). The emission data source is from the German Environment Agency (UBA) for Germany and Thürkow et al. (2024). Detailed data can be referred to Table S12.

*R2-C2: In general, I found it odd that gas phase concentrations are being reported in ug/m3 rather than ppb. Moreover, the units used throughout the paper are broadly inconsistent with production rates being reported in ppb/h, but concentrations reported in ug/m3. I am more familiar with O3/ NOx/ VOC concentrations being reported in ppbv than ug/m3 given that U.S. standards are in those units. But, regardless of the authors preference for units, at minimum the units of the concentrations and production rates should be consistent throughout the paper. For example, in Figure 10, NOx concentrations are in ppb and O3 production is in ppb/h, but in Figure 8 & 9 NOx and O3 concentrations are ug/m3).*

**A2:** We agree the use of different units is confusing. In Germany and Europe, the standards of O$_3$ and NO$_x$ are in µg m$^{-3}$ and all observations are reported in this unit. We therefore chose to discuss the trends of observed concentrations in this unit. Figs. 8 & 9 compare the measured values with our modelled ones and to better link these Figures to the previous observation-based data, we preferred to keep the µg m$^{-3}$ unit here.

For the isopleth plots as in Fig. 10, however, it would be very uncommon to present the production rates in µg h$^{-1}$. It would also be very impractical to convert TNMVOC to µg m$^{-3}$as all the modelling is done on the basis of number density (molec cm$^{-3}$). We therefore prefer to keep these Figures in ppb. To improve the comparability of Fig. 10 with the remaining ones of the manuscript, we now include the conversion factors in the Figure caption, which are also used in the reporting of measurements from the monitoring network.

Figure 10: Isopleth plots for the net O$_3$ production rate (NetPO$_3$ in ppb h$^{-1}$) during 12:00 - 13:00 CET as a function of averaged NO$_x$ and TNMVOC concentrations (A and B for summer and winter cases, respectively). NetPO$_3$ is shown using a rainbow color scale. The coloured points represent the conditions in different years with squares showing traffic stations and triangles showing mean urban stations conditions. The red dotted arrows represent hypothetical future scenarios with reductions in only one precursor (either TNMVOC or NO$_x$), while the yellow dotted arrows illustrate more plausible future pathways, where NetPO$_3$ slightly decreases through strong VOCs emission controls combined with moderate NO$_x$ reductions. Note that based on the underlying modelling, these data in Fig. 10 are given as mixing ratios. For better comparability with mass-based

concentrations elsewhere in the manuscript, conversion factors of $NO_x$ (µg m$^{-3}$) $\approx 1.5 \times NO_x$ (ppb) and $O_3$ (µg m$^{-3}$) $\approx 2 \times O_3$ (ppb) can be used.

***R2-C3:** R/e other reviewer's comments: Overall, I concur with most of their comments and technical notes, particularly in regards to their suggestions to improve the match between figure captions/the text and their detailed notes about things that are missing distinctions within the text. I only note I have one distinct difference in opinion from their comments. References to "p" in the figures is clearly meant to be a p-value indicating statistical significance and to me including it next to the r2 value makes this abundantly clear, as well as the fact that there would be no reason this should be "pressure". However, defining this the first time they reference the p-values in the text would be useful.*

**A3:** Please see our responses to R2-C1 for details on improving the match between figure captions/the text and their detailed notes in the text. To avoid the "missing" information caused by referencing a figure before it is discussed much further down in the text, the relevant details have also been included in the caption of Fig. 10.

p-values means probability. Now "(probability)" has been added in the line 127 and the sentence reads in the manuscript:

Theil-Sen function derives p-values (probability) and uncertainties by bootstrap simulations.

***Specific Comments:***

***R2-C4:** Lines 130-140: This section is missing some details that would be required to reproduce the simulations (e.g. Which version of CAPRAM is used in the work? What meteorology fields/versions are used? What other model options (e.g. specific deposition schemes, emissions inventories etc.) were assumed?) I'm not a CAPRAM user, so I don't know what details are typically provided, but it seems to me that versions/meteorological field names are certainly relevant/ required to reproduce the work.*

**A4:** In the present study, only the detailed near-explicit gas-phase chemistry mechanism MCMv3.3.1 (Master Chemical Mechanism) is applied (this information can be found in the lines 138-139 of the revised manuscript). No version of CAPRAM (Chemical Aqueous Phase Radical Mechanism) is used. Although the SPACCIM model framework allows for the coupling of MCM with CAPRAM to simulate multiphase chemistry, it also enables the use of each mechanism independently. Here, SPACCIM is employed solely with the gas-phase chemistry mechanism MCMv3.3.1.

To avoid misunderstanding, the revised paragraph describing the mechanisms used in lines 132-139 of the text is as follows: To understand the role of photochemistry for $O_3$ concentration evolution in Saxony, photochemical simulations were performed with the air parcel box model SPACCIM (SPectral Aerosol Cloud Chemistry Interaction Model). SPACCIM combines a size-resolved multiphase chemistry model with a microphysical model, enabling both to function independently while accounting for their interdependencies. Detailed descriptions of the SPACCIM model framework can be found in Wolke et al. (2005). In the present study, only the detailed near-explicit gas-phase chemistry mechanism, MCMv3.3.1, is used, which comprises 17224 reactions (http://mcm.york.ac.uk/MCM/) (Saunders et al., 2003).

SPACCIM operates as a box model (fixed at a certain location), where air properties such as temperature, pressure, and relative humidity etc., are defined using fixed parameters based on observed condition to simulate atmospheric chemical processes.

Some specific information on meteorology has been included in lines 149-152 of the article, and some sentences have been revised in lines 146-152. The sentences now read:

Besides emission values, other initial parameters had to be adjusted to their typical daytime and nighttime levels under rural conditions (see Table S4 for details). For meteorological parameters and trace gas concentrations (except $SO_2$, HONO and PAN), data was derived from measurements described in Sect. 2.1. The air temperature was set to 15°C in summer and 4°C in

winter. The pressure was kept constant at 1000 hPa for both seasons, while the relative humidity was maintained at 70%. Ratio of solar radiation, defined as the mean value between 10:00 and 13:00 divided by the maximum clear sky radiation value during the same period, was calculated to 0.7 in summer and 0.4 in winter.

The emission inventories have been described in the lines 143-146 of the manuscript. Also, the setting of dry deposition velocities has been illustrated in the lines of 158-164 of the text.

***R2-C5:*** *Lines 156-157: The choices for the boundary layer heights ascribed need to be justified. These seem reasonable to me, but given the model-measurement discrepancies for NOx/O3 ascribed to this choice, I think its important they state why they were chosen (e.g. match available measurements decently well?). I have more typically seen BLH set in box modeling by toggling them until a secondary species with a long lifetime expected to be lost primarily to dilution is matched.*

**A5:** First, the selected boundary layer heights are based on previously available measurements in Germany during summer and winter (Wiegner et al., 2006; Brümmer et al., 2012; Kotthaus et al., 2023). Then, by scaling the daytime boundary layer heights, a more reasonable agreement is achieved between the modelled $O_3$ concentrations (with BLHs = 2000 m) and the measured values, as shown in the following figure.

[Figure]

The phrase of "based on previously available measurements in Germany (Wiegner et al., 2006; Brümmer et al., 2012; Kotthaus et al., 2023)" has been added at lines of 166-167 of the manuscript to support the selection of boundary layer heights. Now the sentence reads:

The BLHs in the simulations were set to 500 m at night and 2000 m during daytime in summer and to 250 m at night and 1000 m during daytime in winter, based on previously available measurements in Germany (Wiegner et al., 2006; Brümmer et al., 2012; Kotthaus et al., 2023).

Additionally, the references (Wiegner et al., 2006; Brümmer et al., 2012; Kotthaus et al., 2023) have been inserted in lines 911-913, 718-719, and 818-821 of the manuscript. The references now read:

Wiegner, M., Emeis, S., Freudenthaler, V., Heese, B., Junkermann, W., Münkel, C., Schäfer, K., Seefeldner, M., and Vogt, S.: Mixing layer height over Munich, Germany: Variability and comparisons of different methodologies, Journal of Geophysical Research: Atmospheres, 111, 2006.

Brümmer, B., Lange, I., and Konow, H.: Atmospheric boundary layer measurements at the 280 m high Hamburg weather mast 1995-2011. Mean annual and diurnal cycles, Meteorologische Zeitschrift (Berlin), 21, 2012.

Kotthaus, S., Bravo-Aranda, J. A., Collaud Coen, M., Guerrero-Rascado, J. L., Costa, M. J., Cimini, D., O'Connor, E. J., Hervo, M., Alados-Arboledas, L., Jiménez-Portaz, M., Mona, L., Ruffieux, D., Illingworth, A., and Haeffelin, M.: Atmospheric boundary layer height from ground-based remote sensing: a review of capabilities and limitations, Atmos. Meas. Tech., 16, 433-479, 10.5194/amt-16-433-2023, 2023.

***R2-C6:*** *Section 3.3.1 - Were all modeled [NO] concentrations adjusted up by 1 ug/m3 or only nighttime values?? Seems like it's a bit misleading if all modeled [NO] is arbitrarily adjusted up when only the values < DL are adjusted in that manner for the observations.*

**A6:** We appreciate the reviewers' question as it made us realise that, indeed, all simulated NO values were accidentally increased by 1 µg m$^{-3}$ for plotting the data in Fig. 8. As this is obviously wrong, we now provide an updated figure without such increase of simulated NO values and instead remove all observation data below the DL of 1 µg m$^{-3}$, as their concentration cannot be known and were just arbitrarily set to the DL of 1 in the previous figure.

Since the "average of all sites" line was added to the updated Fig. 8 in response to R2-C7, the revised figure and its caption are now as follows:

[Figure]

Figure 8: Diurnal profiles of hourly averaged modelled and observed NO and $NO_2$ in Saxony at rural background sites (A, C for summer case, and B, D for winter cases). Shaded areas indicate night-time, black lines indicate the modelled values, gray lines represent averaged observed concentrations and coloured lines refer to the observed concentrations at each station. It is noted that all observed NO concentrations below the DL of 1 µg m$^{-3}$ are not known and cannot be shown.

To enhance clarity, the paragraph of comparing measured NO and modelled NO at lines 458-475 of the manuscript now reads as follows:

As a first step in model development, the performance of the base cases simulations for Saxony in 2019 was assessed. The year 2019 was chosen because of the best availability of the required input parameters. As a first set of results, the modelled NO and $NO_2$ diurnal patterns during summer and winter are compared with observed data at rural background stations in Fig. 8. It is noted that all measured NO concentrations below the detection limit (DL) of 1 µg m-3 is unknown and therefore not shown in the Figure. Accordingly, the comparison of the NO patterns is only possible for hours with measured NO above the DL. The modelled NO concentrations during the period from 06:00 to 11:00 (CET) demonstrate satisfactory agreement with the observed summer data (Fig. 8A), with the exception of the most polluted Radebeul-Wahnsdorf site. During winter (Fig.

8B), the model seems to underpredict the NO concentrations maybe because of a too high mixing layer being considered. However, the temporal pattern agrees rather well from 09:00-16:00 (CET). Therefore, the linear correlation coefficient (r) exceeds 0.8 from 06:00 to 11:00 in summer and remains above 0.7 from 09:00-16:00 (CET) in winter, indicating a robust correspondence between model outputs and measurements for both seasons. Additionally, the normalized mean bias factor (NMBF) (see its definition in the Supporting Information) (Jaidan et al., 2018) for model-measurement comparisons of NO is -0.03 during 06:00-11:00 in summer and -0.70 during 09:00-16:00 (CET) in winter, indicating satisfactory agreement of model with mean measurements.

**R2-C7:** *Figures 8/9: It would be nice to add an "average of all sites" line that's directly comparable to the single model line in Figures 8 and 9. And would also enable you to calculate the normalized mean bias factor (NMBF) summarizing the model-measurement average disagreement (in addition to r2) between average of all sites and modeled results to more robustly support the conclusions of this section.*

A7: We appreciate this suggestion and several lines of "measured averaged values of all sites" have now been added into the updated Figs. 8 and 9.

See the updated Fig. 8 (see R2-C6) and Fig. 9 and its caption below:

[Figure]

Figure 9: Diurnal profiles of hourly averaged modelled and observed $O_3$ in Saxony at rural background sites (A and B for summer and winter cases, respectively). Shaded areas indicate night-time, black lines indicate the modelled $O_3$, gray lines represent averaged observed $O_3$ and other coloured lines refer to the observed $O_3$ at each station.

In addition, we have now calculated the normalized mean bias factor (NMBF) summarizing the model-measurement average disagreement. The NMBF values for model-measurement comparisons of $O_3$, NO, and $NO_2$ are summarized in the table below.

|  | Normalized mean bias factor (NMBF) | |
|---|---|---|
|  | **Summer** | **Winter** |
| **$O_3$** | 0.12 | -0.16 |
| **NO** | -0.03 during 06:00-11:00 | -0.70 during 09:00-16:00 |
| **$NO_2$** | -0.15 | -0.29 |

In summary, except for NO in winter, the model predictions for $NO_2$ and $O_3$, in both summer and winter, overpredict or underpredict the observed values by no more than ±30% on average, based on the normalized mean bias factor.

The definition of the normalized mean bias factor (NMBF) has been added in lines 87-93 of the Supporting Information, as shown below.

Definition of the normalized mean bias factor (NMBF) as a model performance evaluation metric

The hourly average concentrations of NO, $O_3$, and $NO_2$ observed ($O$) across rural background sites in Saxony are calculated in the same way as the modelled hourly averages ($M$). To evaluate the model performance for each pollutant, the NMBF is used as a statistical metric (Yu et al., 2006; Jaidan et al., 2018), defined as follows:

$$NMBF = \frac{\sum_{i=1}^{n}(M_i - O_i)}{\sum_{i=1}^{n} O_i}$$

where n is the number of hourly mean values for NO, $O_3$ and $NO_2$.

In addition, two references (Jaidan et al., 2018 and Yu et al., 2006) have been inserted in lines 194-196 and 215-216 of the Supporting Information. The reference now reads:

Jaidan, N., El Amraoui, L., Attié, J.-L., Ricaud, P., and Dulac, F.: Future changes in surface ozone over the Mediterranean Basin in the framework of the Chemistry-Aerosol Mediterranean Experiment (ChArMEx), Atmospheric Chemistry and Physics, 18, 9351-9373, 2018.

Yu, S., Eder, B., Dennis, R., Chu, S. H., and Schwartz, S. E.: New unbiased symmetric metrics for evaluation of air quality models, Atmospheric Science Letters, 7, 26-34, 2006.

In the manuscript, a sentence for illustrating model-measurement comparisons of NO has been added in lines 472-475. See the sentences below.

Additionally, the normalized mean bias factor (NMBF) (see its definition in the Supporting Information) (Jaidan et al., 2018) for model-measurement comparisons of NO is -0.03 during 06:00-11:00 in summer and -0.70 during 09:00-16:00 (CET) in winter, indicating satisfactory agreement of model with mean measurements.

A sentence for illustrating model-measurement comparisons of $NO_2$ has been added in lines 490-491 of manuscript. See the sentence below.

Besides, NMBF values for model-measurement comparisons of $NO_2$ is -0.15 in summer and -0.29 in winter, indicating good agreement of model with mean measurements.

A sentence for illustrating model-measurement comparisons of $O_3$ has been added in the lines 498-499 of manuscript. See the sentence below.

Additionally, NMBF values for model-measurement comparisons of $O_3$ is 0.12 in summer and -0.16 in winter, indicating good model performance.

***R2-C8:*** *Figure 10- The markers depicting the years are hard to see on the figure because the lines marking the trends and future emission scenarios are on top of them. I'd suggest flipping the order these are plotted in so that the markers appear on top of the lines so that where each year appears on the figure is easier to see. Additionally. I can't tell the difference between the dark blue that 2010 is plotted in vs the black that 2019 is plotted in, particularly for the summer urban cases that are the focus of the paper. Additionally, I think it would be useful to lower the y-limit a bit on these figures so that it is easier to see what's happening on the lower end of the figures (e.g. whether the reductions in NOx vs. VOCs lead to more /less P(O3) in the future emission scenarios shown as dotted lines on the figure). Some simple edits to this figure would make it much more*

*readable. The caption needs to better describe what the future scenario lines are. These are only mentioned \*much\* further down in the text far after this figure is first referenced. Additionally, the lines don't all appear to be aligned properly to the last 2019 value… Finally, I wonder if its even necessary to "label" these future scenario lines on the figure itself, as these are pretty typical in the literature. Rather, a key describing if it's a NOx/VOC reduction or a reduction of both after 2019 would suffice and clean up the figure a bit.*

A8: The updated Fig. 10 has been incorporated into the manuscript. The modifications include changing the drawing order of lines and points, adjusting the color of the points, slightly lowering the y-axis limit, and removing "label" of future scenario to make the figure cleaner. In addition, the caption of Fig. 10 has been revised.

[Figure]

Figure 10: Isopleth plots for the net $O_3$ production rate (NetPO$_3$ in ppb h$^{-1}$) during 12:00 - 13:00 CET as a function of averaged NO$_x$ and TNMVOC concentrations (A and B for summer and winter cases, respectively). NetPO$_3$ is shown using a rainbow color scale. The coloured points represent the conditions in different years with squares showing traffic stations and triangles showing mean urban stations conditions. The red dotted arrows represent hypothetical future scenarios with reductions in only one precursor (either TNMVOC or NO$_x$), while the yellow dotted arrows illustrate more plausible future pathways, where NetPO$_3$ slightly decreases through strong VOCs emission controls combined with moderate NO$_x$ reductions. Note that based on the underlying modelling, these data in Fig. 10 are given as mixing ratios. For better comparability with mass-based concentrations elsewhere in the manuscript, conversion factors of NO$_x$ (µg m$^{-3}$) ≈ 1.5 × NO$_x$ (ppb) and O$_3$ (µg m$^{-3}$) ≈ 2 × O$_3$ (ppb) can be used.

In addition, to avoid the "missing" information caused by referencing a figure before it is discussed much further down in the text (see the comments in R2-C1 and R2-C3), the relevant details have also been included in the caption of Fig. 10 above. The future scenario pathways were also documented in lines 649-652 of the manuscript. The revised sentences now read as follows:

However, a scenario that only reduce VOCs (as indicated by red dotted arrows to the left in Fig. 10) is not realistic as it comes with the cost of constant NO$_x$. Therefore, a scenario (as indicated by yellow dotted arrows in Fig. 10) where NetPO$_3$ slightly decreases, achieved through strong VOCs emission control with moderate NO$_x$ emission decrease, is more realistic.

***R2-C9:*** *Lines 490-499: I can certainly appreciate the challenge of not having enough VOC data to create Figure 10. But, I'm generally confused about how TNMVOC is estimated here and further clarification is needed in this section especially. The main text states that "The grid of modelled NetPO3 as a function of modelled, inventory-derived NOx and TNMVOC concentrations (see Sect. 2.3), was therefore interpolated to derive TNMVOC concentrations for given measured NOx and dO3/dt. I read this to mean that they used the model with inventory estimates for NOx and TNMVOC which is what is shown in the column for Table S8? And because they see decent agreement between the model & measurements of NetPO3 and NOx*

*that they are assuming TNMVOCs predicted by the model are "right" at each station. What is extremely unclear to me is this "interpolated to derive TNMVOC concentrations for given measured NOx and dO3/dt" statement. That implies that they preformed some sort of correlation/interpolation that is not shown anywhere in the manuscript or supplement to estimate "where to place the markers on Figure 10 on the TNMVOC axis" / get the values shown in Table S9. If that's what was done, I would like to see this figure and have it described with quantitative supporting statistics to convince us that where their TNMVOC estimates are accurate (e.g. show Table S8 as a figure and the equation used to generate the values in Table S9). Additionally, if this methodology has been used in the past for such analysis, it would benefit the paper to reference that such a methodology has been used before in this section to justify this methodology choice. I have seen prior papers use the correlation between CO or HCHO with TNMVOCs to estimate TNMVOC when only CO or HCHO is available in the past, but all of those showed this in the supplement with supporting statistics to show it was a reasonable way to estimate TNMVOCs. Regardless, I'm confused enough about this section that I really don't understand how the values in Table S8 correspond to that shown in Figure S6 or how what's shown in Table S8 is used to get the data in Table S9/ the values used on the TNMVOC axis in Figure 10 and the authors certainly need to clarify this.*

A9: We appreciate the reviewer's valuable comment, which helped clarify the explanation for estimating TNMVOC.

First, the sensitivity simulations for drawing $O_3$ isopleths were done by scaling the base case emissions of TNMVOC and $NO_x$ 20 times in each of three batch runs, i.e. each combination is considered (Table S6 (former Table S5)). Three batches were performed to achieve a sensible range of resulting TNMVOC and $NO_x$ concentrations in the total of 800 and 1200 model runs for summer and winter, respectively. See the newly added Fig. S6 below showing points distribution of resulting TNMVOC and $NO_x$ concentrations.

[Figure]

Figure S6: The distribution of resulting TNMVOC and $NO_x$ concentrations (in ppb) based on a total of 800 and 1200 model runs (see Table S6 for details) for summer (A) and winter (B), respectively. Panels C (summer) and D (winter) are subsets of Panels A and B, respectively, and use the same concentration coordinate ranges as Fig. S7. The different colors indicate different batch runs.

Besides, the averaged instantaneous rate of net ozone production ($NetPO_3$) during noon time of 12:00 - 13:00 CET for each simulated scenario in both summer and winter conditions was obtained for each run. The resulting $NetPO_3$ was interpolated onto a regular 1000 × 1000 grid in the TNMVOC vs. $NO_x$ space to generate Fig. S7 (see below). The $O_3$ isopleths (Fig. 10) were then fitted to this high-resolution grid from Fig. S7. At the same time, two tables present summer and winter data obtained after interpolation. From those, one can identify similar $NO_x$ values along with their corresponding $NetPO_3$ and TNMVOC concentrations.

In our current study, we found in Fig. S8 (former Fig. S6) an excellent correlation between the measured $dO_3/dt$ (especially from 06:00 to 12:00 in summer and 08:00 to 12:00 in winter) and the modeled $NetPO_3$. So, we regard measured $dO_3/dt$ (from 06:00 to 12:00 in summer and 08:00 to 12:00 in winter) a good proxy to the value of modeled $NetPO_3$. By picking the known $NO_x$ and $dO_3/dt$ (finding the close values of $NetPO_3$), the TNMVOC concentration is then identified. The derived TNMVOC together with a comparison of measured and modelled $NO_x$ and $dO_3/dt$ for the station types are given in Table S9 (former S8) and Table S10 (former S9).

[Figure]

Figure S7: The grid of modelled net $O_3$ production rate (NetPO$_3$ in ppb h$^{-1}$) during 12:00 - 13:00 CET as a function of both modelled and emission inventory-derived NO$_x$ and TNMVOC concentrations (in ppb) (see Sect. 2.3) after bivariate linear interpolation. The X-axis and Y-axis in both summer (A) and winter (B) are interpolated using a very fine-resolved 1000 × 1000 grid. NetPO$_3$ is shown using a rainbow color scale.

The revised paragraphs on how to estimate TNMVOC in lines 539 - 557 of the manuscript are as follows:

In the next step, the measured values of NO$_x$ and observed O$_3$ change rate (dO$_3$/dt) for the years 2000, 2005, 2010, 2015, and 2019 were used to indicate for each year the location in the isopleth diagram. A key challenge, however, was the lack of measured TNMVOC concentrations in all years. To work around this, three simulation batches (see Sect.2.3 for details) were performed to achieve a sensible range of resulting TNMVOC and NO$_x$ concentrations in the total of 800 and 1200 model runs for summer and winter, respectively. Points distribution of resulting TNMVOC and NO$_x$ concentrations is shown in Fig. S6. Besides, the averaged NetPO$_3$ during noon time of 12:00 - 13:00 CET for each simulated scenario in both summer and winter conditions was obtained for each run. The resulting NetPO$_3$ was interpolated onto a regular 1000 × 1000 grid in the TNMVOC vs. NO$_x$ space to generate Fig. S7. The O$_3$ isopleths (Fig. 10) were then fitted to this high-resolution grid from Fig. S7. At the same time, two tables present summer and winter data obtained after interpolation. From those, one can identify similar NO$_x$ values along with their corresponding NetPO$_3$ and TNMVOC concentrations.

As depicted in Fig. S8, measured dO$_3$/dt and modelled NetPO$_3$ agreed reasonably well, particularly from 06:00 to 12:00 in summer and 08:00 to 12:00 in winter. This indicates that the measured dO$_3$/dt during these periods serves as a good proxy to the value of modelled NetPO$_3$, which is why it is considered valid to interchange them in the present application. By picking the known NO$_x$ and dO$_3$/dt (finding the close values of NetPO$_3$), the TNMVOC concentration is then identified. For further clarification, in Tables S9 and S10, these TNMVOC estimates are shown together with a comparison of measured and modelled NO$_x$ and dO$_3$/dt for the station types.

As a result of adding two new figures (as Fig. S6 and S7), the numbering of other figures (from Fig. S6) in the Supporting Information has changed accordingly.

**R2-C10:** *Line 499-501: "... because of lowered emissions through environmental mitigations." Are you talking about reductions in anthropogenic VOCs through emission controls/regulations or reductions in biogenic VOCs? Would be useful to clarify what is meant by "environmental mitigations" here / which VOC category is assumed to be affected.*

A10: We were referring to anthropogenic VOCs reductions, yes. The revised sentence in the lines 558-560 of the manuscript now reads:

Notably, simulated TNMVOC concentrations, primarily emitted or locally photochemical in origin, are expected to be similar to or lower than previous measurements (Knobloch et al., 1997) because of lowered anthropogenic emissions through existing European regulation and corresponding mitigation measures.

*R2-C11: Table S4. I appreciate the authors giving the smiles strings of the VOC compounds, but it would be useful for reproducibility if they also supplied the MCM species names of each in another column. As an MCM box modeler, if I wanted to recreate this study, I would have to go through the MCM mechanism and identify each one of these compounds by hand in order to recreate their simulation. Thus, while giving the strings does mean that these concentrations could be used in other mechanisms (and why it is important to retain that information), for reproducibility, it would also be useful to have the MCM compound name (since there are indeed existing "mapping" tools to map those tracer names to the tracers of other mechanisms).*

**A11:** Some SMILES strings are available on the MCM website, while others are not. It is challenging to provide the name of each MCM species. Additionally, there is no publicly accessible webpage in the CAPRAM link for referencing these SMILES strings - they are only accessible to CAPRAM users.

Instead of listing the SMILSE strings twice, one column with the full SMILES strings, one column with their IUPAC names and one column with their chemical structures are presented in the updated table.

As a result of adding one new table of Table S3, the numbering of Table S4 now should be Table S5. The revised caption of Table S5 has been added in line 38 of the Supporting Information, now reads:

See below the revised Table S4.

Table S5. Dominant initial gas-phase concentrations applied in the final 24-hour simulations for summer and winter scenarios. Each species is given by its SMILES string, and its IUPAC name generated using the PubChemPy package in Python (https://pubchempy.readthedocs.io/en/latest/), along with the corresponding chemical structure. It is noted that for some compounds, IUPAC names could not be retrieved from the PubChem database by the tool and they are therefore left blank.

See attached Excel file named *Table S5_Dominant initial gas-phase concentrations.*